# Food additive emulsifiers and cancer risk: Results from the French prospective NutriNet-Santé cohort

**Laury Sellem**[1,2‡], **Bernard Srour**[1,2‡]*, **Guillaume Javaux**[1], **Eloi Chazelas**[1,2], **Benoit Chassaing**[2,3], **Emilie Viennois**[4], **Charlotte Debras**[1,2], **Nathalie Druesne-Pecollo**[1,2], **Younes Esseddik**[1], **Fabien Szabo de Edelenyi**[1], **Nathalie Arnault**[1], **Cédric Agaësse**[1], **Alexandre De Sa**[1], **Rebecca Lutchia**[1], **Inge Huybrechts**[5], **Augustin Scalbert**[5], **Fabrice Pierre**[2,6], **Xavier Coumoul**[2,7], **Chantal Julia**[1,8], **Emmanuelle Kesse-Guyot**[1,2], **Benjamin Allès**[1], **Pilar Galan**[1,2], **Serge Hercberg**[1,2,8], **Mélanie Deschasaux-Tanguy**[1,2], **Mathilde Touvier**[1,2]

**1** Université Sorbonne Paris Nord and Université Paris Cité, INSERM, INRAE, CNAM, Center of Research in Epidemiology and StatisticS (CRESS), Nutritional Epidemiology Research Team (EREN), Bobigny, France, **2** Nutrition And Cancer Research Network (NACRe Network), Jouy-en-Josas, France, **3** INSERM U1016, team "*Mucosal microbiota in chronic inflammatory diseases*", CNRS UMR 8104, Université Paris Cité, Paris, France, **4** INSERM U1149, Center of Research on Inflammation, Université Paris Cité, Paris, France, **5** International Agency for Research on Cancer, World Health Organization, Lyon, France, **6** Toxalim (Research Centre in Food Toxicology), Université de Toulouse, INRAE, ENVT, INP-Purpan, UPS, Toulouse, France, **7** INSERM UMR-S 1124, Université Paris Cité, Paris, France, **8** Public Health Department, Groupe Hospitalier Paris-Seine-Saint-Denis, Assistance Publique-hôpitaux de Paris (AP-HP), Bobigny, France

‡ These authors contributed equally to this work as joint first authors.
* b.srour@eren.smbh.univ-paris13.fr

**Data Availability Statement:** Raw data described in the manuscript are protected and are not available due to data privacy laws according to French regulations. Data can be made available

## Abstract

### Background

Emulsifiers are widely used food additives in industrially processed foods to improve texture and enhance shelf-life. Experimental research suggests deleterious effects of emulsifiers on the intestinal microbiota and the metabolome, leading to chronic inflammation and increasing susceptibility to carcinogenesis. However, human epidemiological evidence investigating their association with cancer is nonexistent. This study aimed to assess associations between food additive emulsifiers and cancer risk in a large population-based prospective cohort.

### Methods and findings

This study included 92,000 adults of the French NutriNet-Santé cohort without prevalent cancer at enrolment (44.5 y [SD: 14.5], 78.8% female, 2009 to 2021). They were followed for an average of 6.7 years [SD: 2.2]. Food additive emulsifier intakes were estimated for participants who provided at least 3 repeated 24-h dietary records linked to comprehensive, brand-specific food composition databases on food additives. Multivariable Cox regressions were conducted to estimate associations between emulsifiers and cancer incidence. Overall, 2,604 incident cancer cases were diagnosed during follow-up (including 750 breast, 322 prostate, and 207 colorectal cancers). Higher intakes of mono- and diglycerides of fatty

upon request pending application and approval. Researchers from public institutions can submit a collaboration request including information on the institution and a brief description of the project to collaboration@etude-nutrinet-sante.fr. All requests will be reviewed by the steering committee of the NutriNet-Santé study within 8 to 12 weeks. If the collaboration request is accepted, a data access agreement will be necessary and appropriate authorisations from the competent administrative authorities may be needed. In accordance with existing regulations, no personal identification data will be accessible. The NutriNet-Santé food composition database is available in the book "Table de composition des aliments, Etude NutriNet-Santé, Editions Inserm – Economia", ISBN-10: 2717865373 ISBN-13: 978-2717865370.

**Funding:** The NutriNet-Santé study was supported by the following public institutions: Ministère de la Santé, Santé Publique France, Institut National de la Santé et de la Recherche Médicale (INSERM), Institut National de la Recherche pour l'agriculture, l'alimentation et l'environnement (INRAE), Conservatoire National des Arts et Métiers (CNAM), Université de Paris, and University Sorbonne Paris Nord. EC was supported by a Doctoral Funding from University Sorbonne Paris Nord - Galilée Doctoral School. LS and CD were supported by a grant from the French National Cancer Institute (INCa). This project has received funding from the European Research Council (ERC) under the European Union's Horizon 2020 research and innovation program (grant agreement No 864219), the French National Cancer Institute (INCa_14059), the French Ministry of Health (arrêté 29.11.19) and the IdEx Université de Paris (ANR-18-IDEX-0001), and a Bettencourt-Schueller Foundation Research Prize 2021. This project was awarded the NACRe (French network for Nutrition And Cancer Research) Partnership Label. BC's laboratory is supported by a Starting Grant from the European Research Council (ERC) under the European Union's Horizon 2020 research and innovation program (grant agreement No. ERC-2018-StG-804135 INVADERS), and the national program "Microbiote" from INSERM. Researchers were independent from funders. Funders had no role in the study design, the collection, analysis, and interpretation of data, the writing of the report, and the decision to submit the article for publication.

**Competing interests:** The authors have declared that no competing interests exist.

**Abbreviations:** ADI, acceptable daily intak; ANSES, Agence nationale de sécurité sanitaire de l'alimentation de l'environnement et du travail (National Agency for Food Environmental and

acids (FAs) (E471) were associated with higher risks of overall cancer (HR $_{high\ vs.\ low\ category}$ = 1.15; 95% CI [1.04, 1.27], p-trend = 0.01), breast cancer (HR = 1.24; 95% CI [1.03, 1.51], p-trend = 0.04), and prostate cancer (HR = 1.46; 95% CI [1.09, 1.97], p-trend = 0.02). In addition, associations with breast cancer risk were observed for higher intakes of total carrageenans (E407 and E407a) (HR = 1.32; 95% CI [1.09, 1.60], p-trend = 0.009) and carrageenan (E407) (HR = 1.28; 95% CI [1.06, 1.56], p-trend = 0.01). No association was detected between any of the emulsifiers and colorectal cancer risk. Several associations with other emulsifiers were observed but were not robust throughout sensitivity analyses. Main limitations include possible exposure measurement errors in emulsifiers intake and potential residual confounding linked to the observational design.

## Conclusions

In this large prospective cohort, we observed associations between higher intakes of carrageenans and mono- and diglycerides of fatty acids with overall, breast and prostate cancer risk. These results need replication in other populations. They provide new epidemiological evidence on the role of emulsifiers in cancer risk.

## Trial registration

ClinicalTrials.gov NCT03335644.

---

## Author summary

### Why was this study done?

- Emulsifiers are widely used food additives in industrially processed foods to improve texture and enhance shelf-life.

- Experimental in vivo/in vitro research as well as a pilot clinical trial on healthy individuals suggests deleterious effects of food additive emulsifier intake on the intestinal microbiota, metabolome, host inflammation, and susceptibility to carcinogenesis.

- To our knowledge, due to challenges to accurately estimate the exposure to food additive emulsifiers through diet, so far there was no available epidemiological evidence from prospective cohorts on food additive emulsifier intakes in relation to cancer risk.

### What did the researchers do and find?

- This study assessed quantitative exposures to a wide range of food additive emulsifiers in a large prospective cohort of adults.

- Higher intakes of mono- and diglycerides of fatty acids (FAs) (E471), total carrageenans (E407, E407a), and carrageenan (E407) were associated with higher risks of overall, breast, and/or prostate cancers.

Occupational Health Safety); EFSA, European Food Safety Authority; FA, fatty acid; FDR, false discovery rate; GNPD, Global New Products Database; GSFA, Codex General Standard for Food Additives; HR, hazard ratio; INRAE, Institut National de la Recherche pour l'agriculture l'alimentation et l'environnement; IPAQ, International Physical Activity Questionnaire; PAL, physical activity level; PCA, principal component analysis; SD, standard deviation; UPF, ultra-processed food; 95% CI, 95% confidence interval.

## What do these findings mean?

- These results provide important epidemiological insights into the role of emulsifiers on cancer risks, and need to be confirmed in further epidemiological and experimental research.

## Introduction

Ultra-processed foods (UPFs) provide a large proportion of dietary energy intakes, with up to 60% in the United States and the United Kingdom [1], and about 30% in France [2] and throughout Europe [3]. Concerns about such high consumptions of UPF have emerged over the past few years, based on large-scale epidemiological studies which suggested diets rich in UPF may be associated with higher risks of noncommunicable diseases [4,5], such as cancers [6], cardiovascular diseases [7,8], type 2 diabetes [9,10], obesity [4,11], and mortality [12,13].

Most UPF contain food additives, which have been proposed as one of the main possible explanations for the deleterious impact of UPF on health [14]. Among the most commonly used food additives, those with emulsifying and thickening properties (referred to as "emulsifiers" thereafter) are added to UPF to improve texture and lengthen shelf-life [15]. At the molecular level, emulsifiers possess both hydrophilic and hydrophobic properties, which is particularly useful to stabilise food preparations that contain lipids. As a consequence, they can be found in thousands of daily-used processed food items (e.g., chocolate, pastries, but also ready-to-eat fruit, vegetable or legume preparations) [16]. The number of authorised emulsifiers varies in the food chain globally, depending upon local definitions and regulations used, but can range from ≈60 in the European Union (EU) to ≈170 in the United States (US) [15]. Although there is no available estimation of emulsifier use among all food additives used in foods worldwide, a recent descriptive study from the French prospective cohort NutriNet-Santé estimated that 7 of the 10 most consumed food additives among French adults were classified as emulsifiers (i.e., total modified starches, lecithins, xanthan gum, pectins, mono- and diglycerides of fatty acids (FAs), carrageenan, and guar gum) [17].

In Europe, the use of emulsifiers in food manufacturing is regulated by the European Food Safety Authority (EFSA), which evaluated their individual safety for consumption and determined acceptable daily intakes (ADIs). Nonetheless, recent in vitro/in vivo experimental studies suggested detrimental effects of food additive emulsifiers such as alterations to the gut microbiota [18–20] and increased low-grade inflammation [19–22]. Microbiota dysbiosis and chronic inflammation may potentially lead to higher risks of gut diseases (including inflammatory bowel disease), but are also involved in the aetiology of many other chronic diseases, including extra intestinal cancers [23,24]. In addition, a first randomised controlled trial in humans demonstrated that short-term intakes of carboxymethylcellulose (European code: E466) in healthy individuals at supraphysiological doses (15 g/day) rapidly altered intestinal microbiota composition and intestinal metabolites production compared to an additive-free diet [25]. However, the impact of food additive emulsifiers on cancer risk or progression is yet to be elucidated and current knowledge is based on scarce, contrasting evidence from experimental studies on animals [26–28]. To our knowledge, no epidemiological study has investigated the links between exposure to emulsifiers and cancer risk in humans, due to important challenges in accurate and reliable estimation of exposure to additive emulsifiers.

In this context, there is a crucial need for large-scale epidemiological studies to understand the role played by food additive emulsifiers on human health, and particularly their potential long-term impact on noncommunicable diseases such as cancers. In the prospective NutriNet-Santé cohort, which collected detailed information on specific commercial brands of industrial food consumed, we recently estimated the intakes of individual food additives (including emulsifiers), in more than 100,000 French adults [17]. Based on this previous work, the present study aims to assess the associations between exposure to food additive emulsifiers and cancer risk in the NutriNet-Santé prospective cohort.

## Methods

### Study population

This study was based on the prospective NutriNet-Santé e-cohort, launched in May 2009, with an open ongoing enrolment of volunteers and the main objective of investigating the relationships between nutrition and health [29]. Participant are recruited from the general population of French adults (aged >18 years) through vast multimedia campaigns. To enrol, participants are required to create a personal account on the NutriNet-Santé web-based platform (https://etude-nutrinet-sante.fr/). Upon enrolment, participants are invited to complete 5 questionnaires about their dietary intakes (detailed below), health (e.g., personal and family history of disease, prescribed medication), anthropometric data (e.g., height, weight) [30,31], physical activity (validated seven-day assessment via the International Physical Activity Questionnaire [IPAQ]) [32], lifestyle and sociodemographic data (e.g., date of birth, sex, education level, professional occupation, smoking status, number of children) [33]. The NutriNet-Santé study is conducted according to the Declaration of Helsinki guidelines and was approved by the Institutional Review Board of the French Institute for Health and Medical Research (IRB Inserm n˚ 0000388FWA00005831) and the "Commission Nationale de l'Informatique et des Libertés" (CNIL n˚908450/n˚909216). It is registered at clinicaltrials.gov as NCT03335644. Electronic informed consent is obtained from each participant. The NutriNet-Santé study was developed to investigate the relationships between multiple dietary exposures and the incidence of chronic diseases, such as cancer. The general protocol of the cohort, written in 2008 before the beginning of the study, is available online [29]. Regarding food additives specifically, the present work is part of a series of prespecified analyses that are included in a project funded by the European Research Council (https://erc.europa.eu/news-events/magazine/erc-2019-consolidator-grants-examples#ADDITIVES).

### Dietary data collection

Usual dietary intakes were assessed at inclusion and then every 6 months, using a series of 3 non-consecutive web-based 24-h dietary records, randomly assigned over a 2-week period (2 weekdays and 1 weekend day). The web-based questionnaires used in the study (available here https://etude-nutrinet-sante.fr/build/qa/docs/guide.htm#environnement) have been tested and validated against both in-person interviews by trained dietitians [34], and urinary and blood markers [35,36] for the key food groups and nutrients (against plasma beta carotene, vitamin C, and n-3 polyunsaturated fatty acids and urinary protein, potassium and sodium), but not food additives. At all times throughout their assigned dietary record period, participants declared all foods and beverages consumed during main meals and any other eating occasion, and estimated portion sizes either by directly entering the weight consumed in the platform, or by using validated photographs or usual containers [37]. A French food composition database (>3,500 items) [38] was used to estimate daily energy, alcohol, macro- and micro-nutrient intakes, which were calculated as the average from all 24-h dietary records

completed during the first 2 years of follow-up. These estimates included contributions from composite dishes using French recipes validated by food and nutrition professionals. Finally, those that underreported total energy intake ($n$ = 21,423, 16.5%) were identified and excluded based on the method proposed by Black (eMethod A in S1 Appendix) [39], inspired from the original method developed by Goldberg [39]. Several quality control operations were also performed to account for overreporting. Details about underreporting and overreporting are presented in eMethod A (S1 Appendix).

## Emulsifier intakes

Intakes of food additives were quantified based on data provided by the participants' dietary records, in which the commercial brand/name of the industrial products consumed were recorded. The detailed method for quantification of food additive intakes was described previously [17]. Briefly, each food item consumed and reported in a specific dietary record was matched against 3 databases to assess the presence of any food additive: Observatoire de la qualité de l'alimentation (OQALI) [40], a national database whose management has been entrusted to the National Institute of Agricultural and Environment Research (INRAE) and the French food safety authority (Agence Nationale Sécurité Sanitaire de l'Alimentation, de l'environnement et du travail—ANSES) to characterise the quality of the food supply, Open Food Facts, an open collaborative database of food products marketed worldwide [16], and Mintel Global New Products Database (GNPD) [41], an online database of innovative food products in the world. In a second step, the dose of food additive ingested with each food item was estimated based on (i) ad hoc laboratory assays quantifying additives in specific food items ($N$ = 2,677 food-additive pairs analysed); (ii) doses in generic food categories provided by the EFSA; or (iii) generic doses from the Codex General Standard for Food Additives (GSFA) [42] (detailed assessment in eMethod B in S1 Appendix).

Among the food additives quantified from the participants' dietary records, we identified 60 food additives classified as emulsifiers or emulsifying salts in the Codex GFSA database [42] and considered the sum of their intakes as the "total emulsifier" exposure. In addition, individual emulsifiers with similar chemical structures were summed into 8 groups: total phosphates, total lactylates, total polyglycerol esters of FAs, total mono and diglycerides of FAs, total celluloses, total carrageenans, total alginates, and total modified starches.

## Cancer case ascertainment

Participants were invited to declare any major health event on a dedicated interface on the study website, either through the yearly health status questionnaire, through a specific health check-up questionnaire sent out every 6 months, or spontaneously. A physician expert committee validated every major health event after reviewing the participants' hospital records and collecting additional information from the participants' treating physicians or hospitals if necessary. In addition, cohort data from participants was linked to medico-administrative databases from the National Health Insurance (SNIIRAM, authorisation by the Council of State No 2013–175) and data from the French national cause-specific mortality registry (CépiDC) to provide additional information on health events and mortality. The International Classification of Diseases-Clinical Modification codes (ICD-CM, 10th revision) was used to classify cancer cases. In this study, we considered as cases all primary cancers diagnosed between 2 years after enrolment and October 5, 2021, with the exception of basal cell carcinoma of the skin, which was not considered as a cancer case.

## Statistical analyses

This study included participants from the NutriNet-Santé cohort who completed at least three 24-h dietary records during their first 2 years of follow-up (as a proxy for dietary habits) and did not have any prevalent cancer diagnosed at baseline (flowchart of participants presented in Fig 1). Baseline participants' characteristics included anthropometric, socioeconomic, health, and dietary data, and were investigated in the total population and compared between sex-specific categories of total emulsifier intakes using $\chi^2$ tests for categorical variables and analysis of variance (ANOVA) tests for continuous variables.

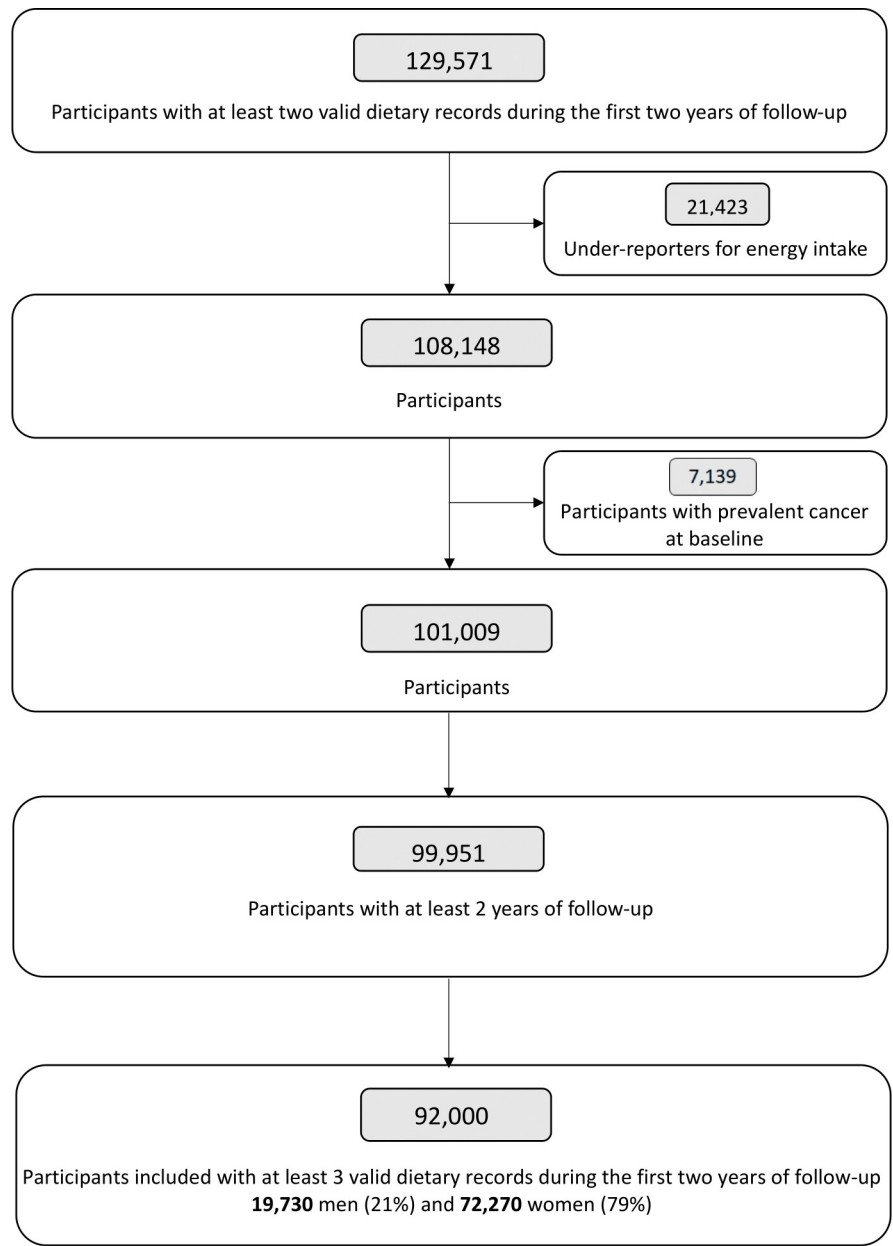

**Fig 1. Flowchart of included participants from the NutriNet-Santé cohort, 2009–2021 ($n$ = 92,000).**

Intakes of emulsifiers were categorised according to sex-specific tertiles into 3 classes: low, medium, and high intakes, except for emulsifiers consumed by less than two thirds of the included participants for which intakes were ranked as non-consumers and 2 levels of intakes (low and high) separated by sex-specific median intakes. The associations between emulsifier intakes and risks of overall, breast, prostate, and colorectal cancers were assessed using multivariable proportional hazard Cox models which computed hazard ratios (HRs) and 95% confidence intervals (95% CI) to compare higher to lower consumers of emulsifiers. P-trends were calculated by handling the categorical variable as an ordinal score. To ensure acceptable statistical power, analyses on individual emulsifiers were restricted to those consumed by at least 5% of the included participants. The proportional hazard assumption was tested using the Schoenfeld residual method (eMethod F in S1 Appendix) [43]. The log-linearity and dose-response relationships between emulsifier intakes and hazard ratios were assessed using restricted cubic splines (eMethod G in S1 Appendix) [44]. Participants contributed person-time to the models from 2 years after their date of enrolment, until the date of cancer diagnosis, the date of death, the date of last completed questionnaire, or October 5, 2021, whichever occurred first. For analyses on premenopausal breast cancer, date of menopause was also considered for censoring. For analyses on postmenopausal breast cancer, follow-up started at date of menopause if participant was included in the cohort prior to her menopause. Cause-specific hazard ratios were computed so that death and cancer events other than the one studied (for site-specific analyses) occurring during follow-up were handled as competing risks, and cumulative incidence functions were calculated using the Fine–Gray subdistribution model (eFigure D in S1 Appendix). Missing values in covariates were handled using multiple imputation by additive regression, bootstrapping, and predictive mean matching ($N = 20$ imputed dataset) as implemented in the *Hmisc* R package (eMethod C in S1 Appendix) [45]. The main model was adjusted for: age (time-scale), sex, body mass index (BMI) (continuous, kg/m$^2$), height (continuous, cm), physical activity (categorical International Physical Activity Questionnaire IPAQ variable: high, moderate, low), smoking status (never smoked, former smoker, current smokers), number of smoked cigarettes in pack-years (continuous), educational level (less than high school degree, <2 y after high school degree, ≥2 y after high school degree), number of dietary records (continuous), family history of cancer (yes/no), energy intake without alcohol (continuous, kcal/d), daily intakes of alcohol (continuous, grams per day (g/d)), lipids (continuous, g/d) (given the lipophilic properties of emulsifiers, and the possible role of lipids in cancer aetiology [46]), sugars (continuous, g/d), sodium (continuous, g/d), fibre (continuous, g/d), consumption levels of fruits and vegetables (continuous, g/d), red and processed meats (continuous, g/d), and dairy products (continuous, g/d). Breast cancer models were additionally adjusted for oral contraception (yes/no, in total and premenopausal models only), age at menarche (never, <12 y, ≥12 y), number of biological children (continuous), age at first biological child (no child, <30 y, ≥30 y), menopausal status at baseline (premenopausal, postmenopausal, in total models only), hormonal treatment for menopause (yes/no, in total and postmenopausal models only). Sensitivity analyses were conducted for all emulsifiers with at least 1 statistically significant association with cancer risk and are fully described in eMethod E in S1 Appendix.

All methods have been described in line with the Strengthening the Reporting of Observational Studies in nutritional Epidemiology guidelines (S1 STROBE Checklist). Multi-adjusted Cox models for several confounders were prespecified. The analyses added following the peer-review process were as follows: restricting the sample to participants with at least 3 dietary records, starting follow-up 2 years after enrolment, further adjustments (for food groups and nutrients instead of dietary patterns, for artificial sweeteners in sensitivity analyses), unadjusting for intakes of other emulsifiers, an analysis using all dietary records available during

follow-up, any versus non comparisons, principal component analysis to compute emulsifier patterns (eMethod D in S1 Appendix), and cumulative incidence functions. All statistical tests were two-sided, and *p*-values <0.05 were considered statistically significant. All statistical analyses were conducted in R version 4.1.2 [47], except from the restricted cubic spline method and the cumulative incidence functions which were implemented in SAS version 9.4.

## Results

### Descriptive characteristics

A total of $N$ = 92,000 participants, among which 78.6% women, were included in this study (Fig 1), with a mean age of 44.5 y (SD 14.5) at baseline (Table 1). Mean number of completed 24-h dietary records was 6.0 (SD 3.1). The distribution of the number of dietary records per study participant is provided in eMethod A (S1 Appendix). A total of 99.8% of participants consumed at least 1 food additive emulsifier. Compared to low-consumers of total emulsifiers, high-consumers were younger, less likely to smoke, with lower alcohol intake and exhibited higher BMI at baseline, higher educational level, physical activity level, dietary energy intake, and proportion of UPF in their diet.

Contributions of individual food additive emulsifiers to intakes of total emulsifiers, absolute intakes of emulsifiers (in mg/d), and correlations between intakes of individual emulsifiers are presented in Fig 2 and Table 2, and eFigure A in S1 Appendix, respectively. A total of 32 individual emulsifiers were consumed by <5% of the included participants and were therefore not studied individually in relation to cancer risk (Table 2). These emulsifiers were, however, included in the calculations of total and groups of emulsifier intakes. Finally, dietary sources of total emulsifiers were varied, with main contributors including processed fruits and vegetables, cakes and biscuits, and dairy products (Fig 3, eTable A). eTable B in S1 Appendix provides mean intakes of each emulsifier in each category of intake, and category cut-offs are provided in footnotes to eTable B in S1 Appendix.

### Associations between emulsifier intakes and cancer risks

Between enrolment + 2 years and 2021, 615,749 person-years contributed to this study, with a mean follow-up of 6.7 y (SD 2.2). In total, 2,604 incident cancer cases were diagnosed, including, for example, 750 breast cancers, 322 prostate cancers, 207 colorectal cancers, 162 melanomas, 124 lung cancers, 110 squamous cell carcinomas, and 90 lymphomas. Information regarding specific cancer subtypes was available for 1,414 cases: the most frequent breast cancer subtypes were oestrogen–positive (ER+) and progesterone–positive (PR+) (85% and 75%, respectively), while triple negative breast cancers represented 10% of all breast cancer cases. At diagnosis, breast cancer was localised, advanced, and metastatic in 69.6%, 28.9%, and 0.2% of cases, respectively. As regards prostate cancer, 42% of cases were low-risk tumours (Gleason score 6 and below), 45% intermediate risk (Gleason score 7), and 13% high-risk tumour (Gleason score 8 and above). Absolute incidence rates according to categories of emulsifier intakes, standardised by age and sex, are presented in eTable C in S1 Appendix.

Overall, Schoenfeld residual plots did not show evidence for violation of the proportional hazard assumptions (eFigure B in S1 Appendix). The main associations between emulsifier intakes and cancer risks are presented in Fig 4, and all tested associations as well as category cut-offs are detailed in eTable D in S1 Appendix.

In the main models, higher intakes of mono- and diglycerides of FAs (E471) were associated with higher risks of overall cancer (HR high vs. low category = 1.15; 95% CI [1.04, 1.27], p-trend = 0.01), breast cancer (HR = 1.24; 95% CI [1.03, 1.51], p-trend = 0.04), and prostate cancer (HR = 1.46; 95% CI [1.09, 1.97], p-trend = 0.02). In addition, associations with breast

**Table 1. Baseline characteristics of study participants from the NutriNet-Santé cohort, 2009–2021 (*N* = 92,000).**

| | | Sex-specific tertiles of total emulsifier intakes[a] | | | |
| --- | --- | --- | --- | --- | --- |
| | Overall | Low intake | Medium intake | High intake | *p*-value[b] |
| **Number of participants** | 92,000 | 30,667 | 30,666 | 30,667 | |
| **Age (years), Mean (SD)** | 44.5 (14.5) | 46.2 (14.7) | 44.9 (14.6) | 42.5 (13.9) | <0.001 |
| **Women, *N* (%)** | 72,270 (78.6) | 24,090 (78.6) | 24,090 (78.6) | 24,090 (78.6) | |
| **Height (cm), Mean (SD)** | 166.7 (8.1) | 166.3 (8.1) | 166.5 (8.1) | 167.3 (8.2) | <0.001 |
| *Missing values, N (%)* | 789 (0.01) | 261 (0.01) | 228 (0.01) | 300 (0.01) | |
| **BMI (kg/m$^2$), Mean (SD)** | 23.7 (4.4) | 23.6 (4.4) | 23.7 (4.4) | 23.8 (4.6) | <0.001 |
| *Missing values, N (%)* | 789 (0.01) | 261 (0.01) | 228 (0.01) | 300 (0.01) | |
| **Family history of cancer, *N* (%)** | 29,679 (32.6) | 10,324 (34.1) | 9,990 (32.9) | 9,365 (30.8) | <0.001 |
| *Missing values, N (%)* | 950 (0.01) | 387 (0.01) | 271 (0.01) | 292 (0.01) | |
| **Education level, *N* (%)** | | | | | <0.001 |
| *Less than high school degree* | 14,917 (16.3) | 5,520 (18.2) | 5,023 (16.5) | 4,374 (14.4) | |
| *<2 years after high school* | 14,172 (15.5) | 4,961 (16.3) | 4,664 (15.3) | 4,547 (14.9) | |
| *≥2 years after high school* | 62,156 (68.1) | 19,911 (65.5) | 20,731 (68.2) | 21,514 (70.7) | |
| *Missing values, N (%)* | 755 (0.01) | 275 (0.01) | 248 (0.01) | 232 (0.01) | |
| **Smoking status, *N* (%)** | | | | | <0.001 |
| *Never* | 41,776 (45.4) | 12,814 (41.8) | 14,068 (45.9) | 14,894 (48.6) | |
| *Former smoker* | 37,500 (40.8) | 13,032 (42.5) | 12,668 (41.3) | 11,800 (38.5) | |
| *Current smoker* | 12,686 (13.8) | 4,805 (15.7) | 3,921 (12.8) | 3,960 (12.9) | |
| *Missing values, N (%)* | 38 (<0.001) | 16 (<0.001) | 9 (<0.001) | 13 (<0.001) | |
| **IPAQ physical activity level, *N* (%)** | | | | | <0.001 |
| *Low* | 25,836 (32.5) | 9,074 (34.4) | 8,693 (32.5) | 8,069 (30.5) | |
| *Moderate* | 34,399 (43.2) | 11,211 (42.5) | 11,527 (43.1) | 11,661 (44.0) | |
| *High* | 19,364 (24.3) | 6,090 (23.1) | 6,527 (24.4) | 6,747 (25.5) | |
| *Missing values, N (%)* | 12,401 (13.5) | 4,292 (14) | 3,919 (12.8) | 4,190 (13.7) | |
| **Number of biological children, Mean (SD)** | 1.3 (1.2) | 1.3 (1.2) | 1.3 (1.2) | 1.2 (1.2) | <0.001 |
| **Baseline menopausal status, *N* (%)** | | | | | <0.001 |
| *Post-menopausal* | 17,679 (24.5) | 6,828 (28.3) | 6,095 (25.3) | 4,756 (19.7) | |
| *Pre-menopausal* | 54,591 (75.5) | 17,262 (71.7) | 17,995 (74.7) | 19,334 (80.3) | |
| **Use of hormonal treatment for menopause, *N* (%)** | 3,204 (3.5) | 1,215 (4) | 1,102 (3.6) | 887 (2.9) | <0.001 |
| **Use of oral contraception, *N* (%)** | 20,384 (22.2) | 5,993 (19.5) | 6,786 (22.1) | 7,605 (24.8) | <0.001 |
| **Energy intake without alcohol (kcal/d), Mean (SD)** | 1,836.4 (443.8) | 1,705.2 (411.7) | 1,824.5 (406.9) | 1,979.5 (466.9) | <0.001 |
| **Alcohol intake (g/d), Mean (SD)** | 7.9 (11.7) | 8.6 (13) | 7.9 (11.4) | 7.2 (10.7) | <0.001 |
| **Total lipid intake (g/d), Mean (SD)** | 81.8 (24.8) | 75.7 (23.5) | 81.3 (23.1) | 88.4 (26.1) | <0.001 |
| **Sodium intake (mg/d), Mean (SD)** | 2,726.1 (870.7) | 2,576.5 (866.1) | 2,738.3 (835.2) | 2,863.5 (886.2) | <0.001 |
| **Fibre intake (g/d), Mean (SD)** | 19.5 (7.1) | 19.3 (7.8) | 19.3 (6.7) | 19.9 (6.8) | <0.001 |
| **Sugar intake (g/d), Mean (SD)** | 92.9 (32.6) | 82.9 (31) | 91.9 (29.3) | 104 (33.8) | <0.001 |
| **Fruit and vegetable intake (g/d), Mean (SD)** | 408.5 (218) | 421.9 (237.1) | 403.1 (205.4) | 400.4 (209.4) | <0.001 |
| **Wholegrain food intake (g/d), Mean (SD)** | 34.4 (45.4) | 38.1 (50.8) | 33.7 (42.7) | 31.3 (42) | <0.001 |
| **Total dairy intake (g/d), Mean (SD)** | 198.1 (147) | 187.8 (148.5) | 199.6 (142.6) | 206.9 (149.2) | <0.001 |
| **Red and processed meat intake (g/d), Mean (SD)** | 101.9 (59.3) | 99.8 (62.6) | 101.8 (57.1) | 104.2 (58) | <0.001 |
| **Ultra-processed food intake (% daily weight intake), Mean (SD)** | 17.2 (9.6) | 14.2 (8.9) | 17.3 (8.9) | 20.2 (10.0) | <0.001 |
| **Total emulsifier intake (mg/d), Mean (SD)** | 4,275.2 (3,080.1) | 1,524.6 (720.5) | 3,687.1 (645) | 7,614 (2,909.7) | <0.001 |

[a] Cut-offs for total emulsifier intakes 2,701.3 and 5,162.5 mg/d in men and 2,618.5 and 4,790.6 mg/d in women.

[b] Obtained from χ2 tests for categorical variables and ANOVA tests for continuous variables.

ANOVA, analysis of variance; BMI, body mass index; IPAQ, International Physical Activity Questionnaire.

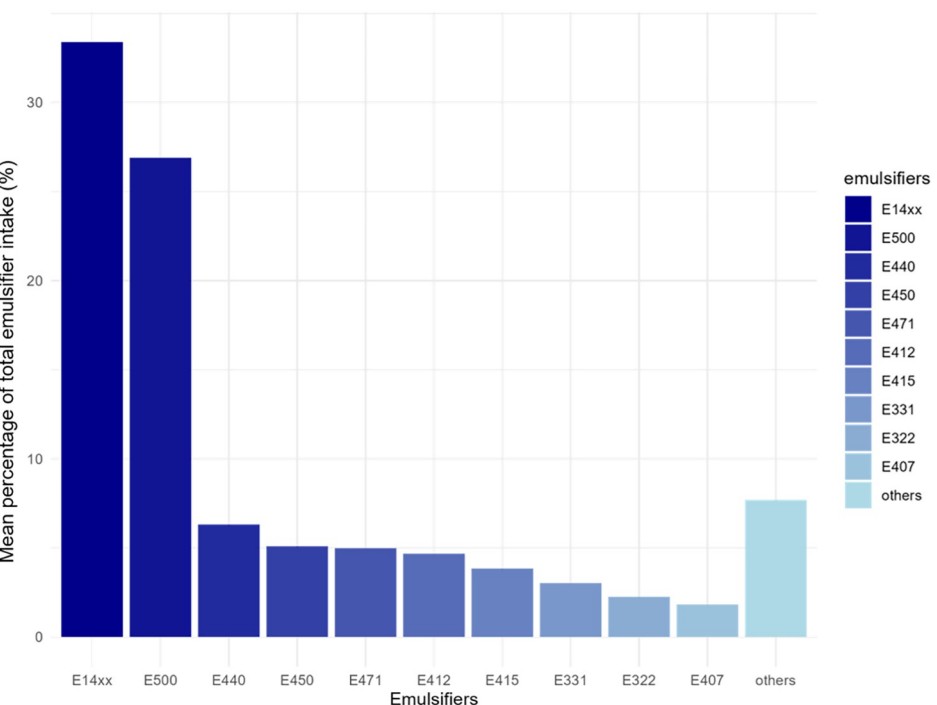

**Fig 2. Dietary sources of total and groups of emulsifier intakes among study participants from the NutriNet-Santé cohort, 2009–2021 (*N* = 92,000).[a,b] FAs, fatty acids.** [a]Groups of emulsifiers were defined as follows (European codes): total phosphates (E339, E340, E341, E343, E450, E451, E452), total lactylates (E481, E482), total polyglycerol esters of FAs (E475, E476), total mono and diglycerides of FAs (E471, E472, E472a, E472b, E472c, E472e), total celluloses (E460, E461, E464, E466, E468), total carrageenans (E407, E407a), total alginates (E400, E401, E402, E404, E405), and total modified starches (E14xx). [b]Detailed % are presented in eTable A in S1 Appendix.

cancer risk were observed for higher intakes of total carrageenans (E407 and E407a) (HR = 1.32; 95% CI [1.09, 1.60], p-trend = 0.009) and carrageenan (E407) (HR = 1.28; 95% CI [1.06, 1.56], p-trend = 0.01). In analyses by menopausal status, higher risks of premenopausal breast cancer were associated with higher intakes of diphosphates (E450) (HR = 1.45; 95% CI [1.04, 2.02], p-trend = 0.03), pectins (E440) (HR = 1.55; 95% CI [1.12, 2.14], p-trend = 0.008), and sodium bicarbonate (E500) (HR = 1.48; 95% CI [1.07, 2.05], p-trend = 0.01). No significant association was detected between studied emulsifiers and colorectal cancer risk (eTable D in S1 Appendix) in our study. Overall, these main results were similar in all sensitivity analyses (eMethod E, eTables E, F and G in S1 Appendix).

In addition, the following associations were observed in the main model but were not robust in all sensitivity analyses, especially when all dietary records available during follow-up (up to 62 records) were used to calculate emulsifier intakes (Fig 4 and eTable E–model 4): (i) higher intakes of Xanthan gum (E415) were associated with a higher risk of overall cancer (HR in the main model = 1.13; 95% CI [1.02, 1.25], p-trend = 0.04); (ii) higher risks of breast cancers were associated with higher intakes of polyglycerol esters of FAs (E475) (HR = 1.50; 95% CI [1.05, 2.15], p-trend = 0.02) and carob bean gum (E410) (HR = 1.19; 95% CI [0.99, 1.43], p-trend = 0.045); (iii) higher intakes of total carrageenans (E407 and E407a) were associated with postmenopausal breast cancer (HR = 1.28; 95% CI [1.00, 1.64], p-trend = 0.04); (iv) higher intakes of dipotassium phosphate (E340) were associated with a higher risk of premenopausal breast cancer (HR = 1.77; 95% CI [1.08, 2.91], p-trend = 0.03); (v) higher risks of prostate cancer were associated with higher intakes of Guar gum (E412) (HR = 1.39; 95% CI [1.04, 1.87],

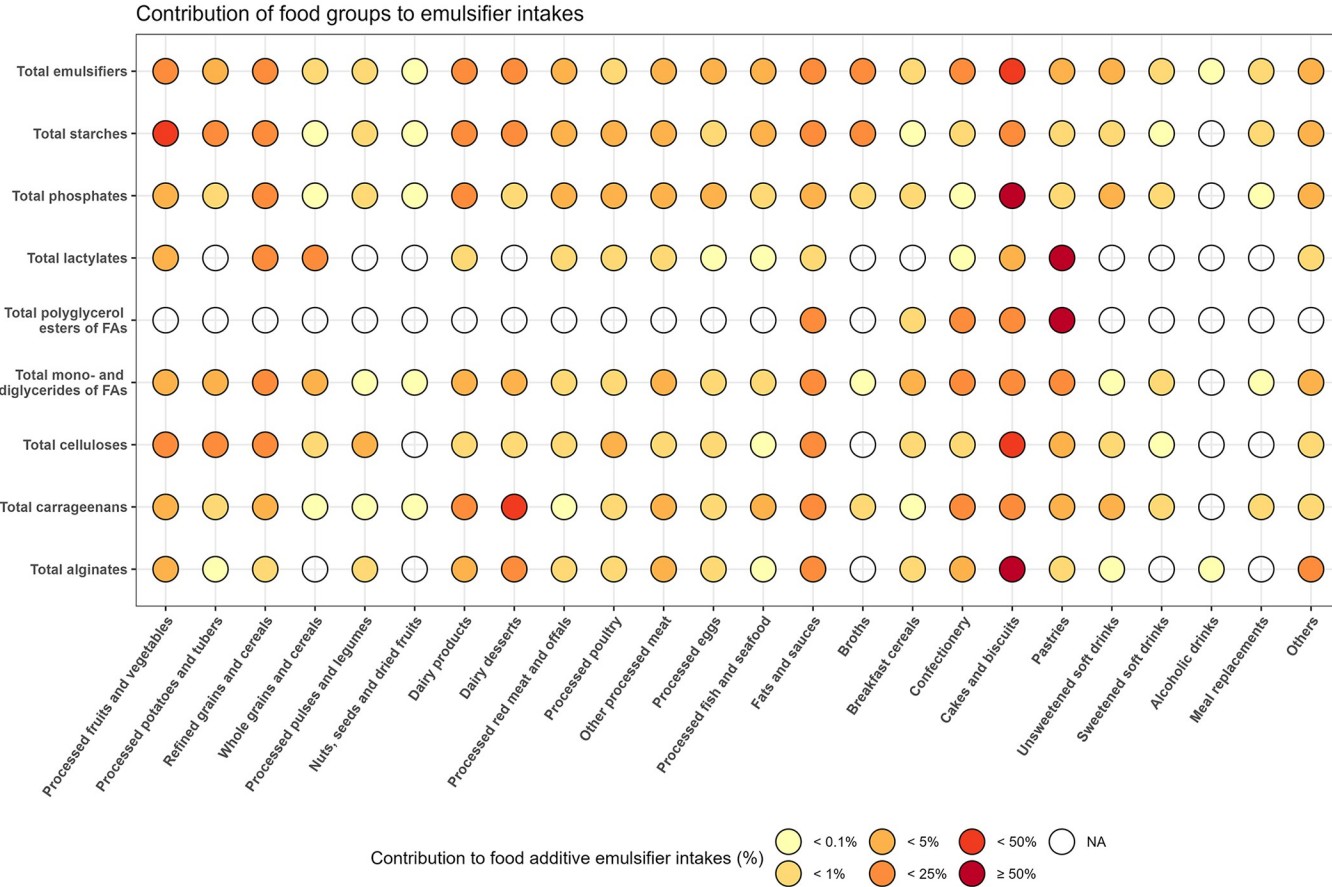

**Fig 3. Contribution of individual emulsifiers to total emulsifier intakes (%) among study participants from the NutriNet-Santé cohort, 2009–2021 (N = 92,000).[a] FAs, fatty acids[a].** Other emulsifiers included (ordered by descending contributions): triphosphates (E451), gum arabic (E414), polyphosphates (E452), carob bean gum (E410), cellulose (E460), tricalcium phosphate (E341), mono and diacetyl tartaric acid esters of mono- and diglycerides of FAs (E472e), hydroxypropyl methyl cellulose (E464), polyglycerol esters of FAs (E475), lactic acid esters of mono- and diglycerides of FAs (E472b), sodium stearoyl-2-lactylate (E481), sodium alginate (E401), ammonium salts of phosphatidic acid (E442), esters of mono- and diglycerides of FAs (E472), polyglycerol esters of interesterified ricinoleic acid (E476), citric acid esters of mono- and diglycerides of FAs (E472c), silicon dioxide (E551), tripotassium phosphate (E340), methyl cellulose (E461), carboxymethylcellulose (E466), trisodium phosphate (E339), acetic acid esters of mono- and diglycerides of FAs (E472a), agar (E406), sucrose esters of FAs (E473), propylene glycol esters of FAs (E477), gellan gum (E418), sorbitan tristearate (E492), processed Euchema seaweed (E407a), beeswax (E901), potassium alginate (E402), maltitol (E965), triethyl citrate (E1505), xylitol (E967), glycerol esters of rosin (E445), polyoxyethylene sorbitan monooleate (E433), potassium dihydrogen citrate (E332), calcium alginate (E404), calcium stearoyl-2-lactylate (E482), konjac flour (E425), cross-linked sodium carboxymethylcellulose (E468), sucrose acetate isobutyrate (E444), sodium tartarate (E335), polyoxyethylene sorbitan monostearate (E435), sorbitan monostearate (E491), alginic acid (E400), propylene glycol (E1520), quillaia extract (E999), sodium aluminium phosphate (E541), magnesium hydrogen phosphate (E343), propylene glycol alginate (E405), and dimethyl polysiloxane (E900).

p-trend = 0.02), gum arabic (E414) (HR = 2.53; 95% CI [1.54, 4.15], p-trend = 0.009), and beeswax (E901) (HR = 2.43; 95% CI [1.14, 5.18], p-trend = 0.03).

In restricted cubic spline curves (eFigure C in S1 Appendix), *p*-values for nonlinearity tests show mostly linear trends. However, for some associations, nonlinear relationships are suggested by the plots (e450, e471, e440, and e500 with premenopausal breast cancer; e412 with prostate cancer; e415 with overall and breast cancer).

## Discussion

In this large prospective cohort study, we observed increased cancer risks associated with higher intakes of 5 individual and 1 group of food additive emulsifiers that are widely used in

**Table 2. Daily emulsifier intakes among study participants from the NutriNet-Santé cohort, 2009–2021 (*N* = 92,000).**

| Emulsifier name | European code | Mean intake (mg/d) | SD | Median intake (mg/d) | 25th percentile (mg/d) | 75th percentile (mg/d) | Proportion of consumers (%) |
|---|---|---|---|---|---|---|---|
| Total emulsifiers | | 4275.2 | 3080.1 | 3651.5 | 2141.2 | 5666.9 | 99.8 |
| **Total alginates** | | 8.8 | 34.0 | 0.0 | 0.0 | 0.0 | 15.7 |
| Alginic acid | E400 | 0.0 | 0.8 | 0.0 | 0.0 | 0.0 | 0.1 |
| Sodium alginate | E401 | 8.5 | 33.5 | 0.0 | 0.0 | 0.0 | 15.0 |
| Potassium alginate | E402 | 0.3 | 4.8 | 0.0 | 0.0 | 0.0 | 0.8 |
| Calcium alginate | E404 | 0.1 | 3.7 | 0.0 | 0.0 | 0.0 | <0.1 |
| Propylene glycol alginate | E405 | 0.0 | 0.0 | 0.0 | 0.0 | 0.0 | <0.1 |
| **Total carrageenans** | | 60.1 | 73.6 | 38.8 | 2.4 | 88.3 | 78.7 |
| Carrageenan | E407 | 57.6 | 71.3 | 37.1 | 1.7 | 84.0 | 77.9 |
| Processed Euchema seaweed | E407a | 2.5 | 13.6 | 0.0 | 0.0 | 0.0 | 9.1 |
| **Total phosphates** | | 360.3 | 492.4 | 229.1 | 44.6 | 497.1 | 79.8 |
| Trisodium phosphate | E339 | 9.1 | 56.4 | 0.0 | 0.0 | 0.0 | 6.1 |
| Tripotassium phosphate | E340 | 8.1 | 92.2 | 0.0 | 0.0 | 0.0 | 5.6 |
| Tricalcium phosphate | E341 | 27.9 | 232.9 | 0.0 | 0.0 | 0.0 | 18.0 |
| Magnesium hydrogen phosphate | E343 | 0.0 | 0.0 | 0.0 | 0.0 | 0.0 | <0.1 |
| Diphosphates | E450 | 246.7 | 341.0 | 141.1 | 0.0 | 342.2 | 72.6 |
| Sodium tripolyphosphate | E451 | 44.4 | 116.5 | 0.0 | 0.0 | 10.3 | 25.7 |
| Polyphosphates | E452 | 24.2 | 80.9 | 0.0 | 0.0 | 0.0 | 22.5 |
| **Total celluloses** | | 18.9 | 92.4 | 0.0 | 0.0 | 0.0 | 20.8 |
| Cellulose | E460 | 9.8 | 69.2 | 0.0 | 0.0 | 0.0 | 10.3 |
| Methyl cellulose | E461 | 1.9 | 16.8 | 0.0 | 0.0 | 0.0 | 2.4 |
| Hydroxypropyl methyl cellulose | E464 | 3.3 | 32.1 | 0.0 | 0.0 | 0.0 | 4.4 |
| Carboxymethylcellulose | E466 | 3.8 | 30.1 | 0.0 | 0.0 | 0.0 | 10.7 |
| Cross-linked sodium carboxymethylcellulose | E468 | 0.0 | 0.1 | 0.0 | 0.0 | 0.0 | 0.1 |
| **Total mono- and diglycerides of FAs** | | 205.6 | 273.5 | 124.8 | 22.9 | 282.1 | 83.9 |
| Mono-and diglycerides of FAs | E471 | 162.3 | 202.2 | 101.1 | 11.0 | 232.9 | 81.5 |
| Esters of mono- and diglycerides of FAs | E472 | 3.3 | 37.7 | 0.0 | 0.0 | 0.0 | 1.3 |
| Acetic acid esters of mono- and diglycerides of FAs | E472a | 5.9 | 75.5 | 0.0 | 0.0 | 0.0 | 3.2 |
| Lactic acid esters of mono- and diglycerides of FAs | E472b | 20.8 | 100.3 | 0.0 | 0.0 | 0.0 | 13.4 |
| Citric acid esters of mono- and diglycerides of FAs | E472c | 8.2 | 53.1 | 0.0 | 0.0 | 0.0 | 7.3 |
| Mono and diacetyl tartaric acid esters of mono- and diglycerides of FAs | E472e | 5.1 | 27.2 | 0.0 | 0.0 | 0.0 | 14.3 |
| **Total polyglycerol esters of FAs** | | 14.4 | 62.7 | 0.0 | 0.0 | 0.0 | 21.7 |
| Polyglycerol esters of FAs | E475 | 10.7 | 60.7 | 0.0 | 0.0 | 0.0 | 7.0 |
| Polyglycerol esters of interesterified ricinoleic acid | E476 | 3.7 | 15.4 | 0.0 | 0.0 | 0.0 | 16.0 |
| **Total lactylates** | | 4.2 | 22.1 | 0.0 | 0.0 | 0.0 | 8.6 |
| Sodium stearoyl-2-lactylate | E481 | 4.1 | 21.7 | 0.0 | 0.0 | 0.0 | 8.5 |
| Calcium stearoyl-2-lactylate | E482 | 0.1 | 3.1 | 0.0 | 0.0 | 0.0 | 0.2 |
| **Total modified starches** | E14xx | 1299.7 | 1116.6 | 1055.3 | 494.7 | 1809.7 | 92.9 |
| Lecithins | E322 | 61.1 | 76.6 | 38.1 | 11.1 | 83.6 | 88.4 |
| Sodium citrate | E331 | 117.6 | 270.4 | 0.0 | 0.0 | 128.6 | 49.6 |
| Potassium dihydrogen citrate | E332 | 0.0 | 0.0 | 0.0 | 0.0 | 0.0 | <0.1 |
| Sodium tartrates | E335 | 0.0 | 0.4 | 0.0 | 0.0 | 0.0 | <0.1 |

(*Continued*)

**Table 2.** (Continued)

| Emulsifier name | European code | Mean intake (mg/d) | SD | Median intake (mg/d) | 25th percentile (mg/d) | 75th percentile (mg/d) | Proportion of consumers (%) |
|---|---|---|---|---|---|---|---|
| Agar | E406 | 3.2 | 34.2 | 0.0 | 0.0 | 0.0 | 2.0 |
| Carob bean gum | E410 | 31.5 | 67.5 | 0.0 | 0.0 | 37.9 | 45.7 |
| Guar gum | E412 | 167.3 | 224.4 | 90.9 | 0.0 | 238.3 | 72.9 |
| Gum arabic | E414 | 53.1 | 407.0 | 0.0 | 0.0 | 0.0 | 10.7 |
| Xanthan gum | E415 | 135.0 | 213.3 | 50.3 | 9.4 | 176.9 | 82.8 |
| Gellan gum | E418 | 0.4 | 4.5 | 0.0 | 0.0 | 0.0 | 2.0 |
| Konjac flour | E425 | 0.0 | 0.9 | 0.0 | 0.0 | 0.0 | <0.1 |
| Polyoxyethylene sorbitan monooleate | E433 | 0.3 | 4.6 | 0.0 | 0.0 | 0.0 | 1.0 |
| Polyoxyethylene sorbitan monostearate | E435 | 0.0 | 0.9 | 0.0 | 0.0 | 0.0 | <0.1 |
| Pectins | E440 | 218.1 | 303.9 | 130.0 | 31.4 | 285.7 | 82.5 |
| Ammonium salts of phosphatidic acid | E442 | 6.2 | 42.5 | 0.0 | 0.0 | 0.0 | 10.6 |
| Sucrose acetate isobutyrate | E444 | 0.0 | 0.8 | 0.0 | 0.0 | 0.0 | 0.1 |
| Glycerol esters of rosin | E445 | 0.1 | 1.2 | 0.0 | 0.0 | 0.0 | 1.8 |
| Sucrose esters of FAs | E473 | 1.2 | 12.8 | 0.0 | 0.0 | 0.0 | 2.6 |
| Propylene glycol esters of FAs | E477 | 0.4 | 7.4 | 0.0 | 0.0 | 0.0 | 1.9 |
| Sorbitan monostearate | E491 | 0.2 | 5.5 | 0.0 | 0.0 | 0.0 | 0.2 |
| Sorbitan tristearate | E492 | 0.6 | 11.6 | 0.0 | 0.0 | 0.0 | 0.5 |
| Sodium bicarbonate | E500 | 1,489.8 | 2,043.8 | 750.0 | 0.0 | 2,163.2 | 74.2 |
| Sodium aluminium phosphate | E541 | 0.0 | 0.0 | 0.0 | 0.0 | 0.0 | <0.1 |
| Silicon dioxide | E551 | 8.0 | 181.0 | 0.0 | 0.0 | 0.0 | 2.6 |
| Dimethyl polysiloxane | E900 | 0.0 | 0.0 | 0.0 | 0.0 | 0.0 | <0.1 |
| Beeswax | E901 | 0.1 | 0.6 | 0.0 | 0.0 | 0.0 | 6.1 |
| Maltitol | E965 | 6.3 | 91.4 | 0.0 | 0.0 | 0.0 | 2.1 |
| Xylitol | E967 | 2.4 | 35.4 | 0.0 | 0.0 | 0.0 | 1.3 |
| Quillaia extract | E999 | 0.0 | 0.1 | 0.0 | 0.0 | 0.0 | <0.1 |
| Triethyl citrate | E1505 | 0.4 | 3.6 | 0.0 | 0.0 | 0.0 | 2.2 |
| Propylene glycol | E1520 | 0.0 | 0.3 | 0.0 | 0.0 | 0.0 | <0.1 |

FAs, fatty acids; SD, standard deviation.

Europe, such as carrageenan (E407), diphosphates (E450), mono- and diglycerides of FAs (E471), pectins (E440) and sodium bicarbonate (E500).

Food additive emulsifiers have been evaluated in recent EFSA reports, which did not conclude in any safety concern or need for a numerical admissible daily intake (ADI) for sodium citrate (E331) [48], carob bean gum (E410) [49], xanthan gum (E415) [50], mono- and diglycerides of FAs (E471) [51], or total celluloses (E460, E461, E464, E466, E468) [52]. Intakes of emulsifiers in our study were lower than those reported in the EFSA opinions using simulation scenarios based on maximum permitted levels, and no brand-specific data [50,52–54], and were of the same order of magnitude as those reported in the American Cancer Prevention Study-3 (CPS-3) Diet Assessment Sub-Study, using brand-specific qualitative data coupled with simulations for quantitative data (4.2 g/day of total emulsifiers in our study, versus 1.9 g/day of total emulsifiers in CPS-3) [22]. In line with data previously published on the NutriNet-Santé prospective cohort study, ADIs for tartaric acid esters of mono- and diglycerides of FAs (E472e, set at 240 mg/kg of body weight/d) [55], for total lactylates (E481 and E482, set at 22 mg/kg of body weight/d) [56], for carrageenan (E407, set at 75 mg/kg of body weight/d) [53],

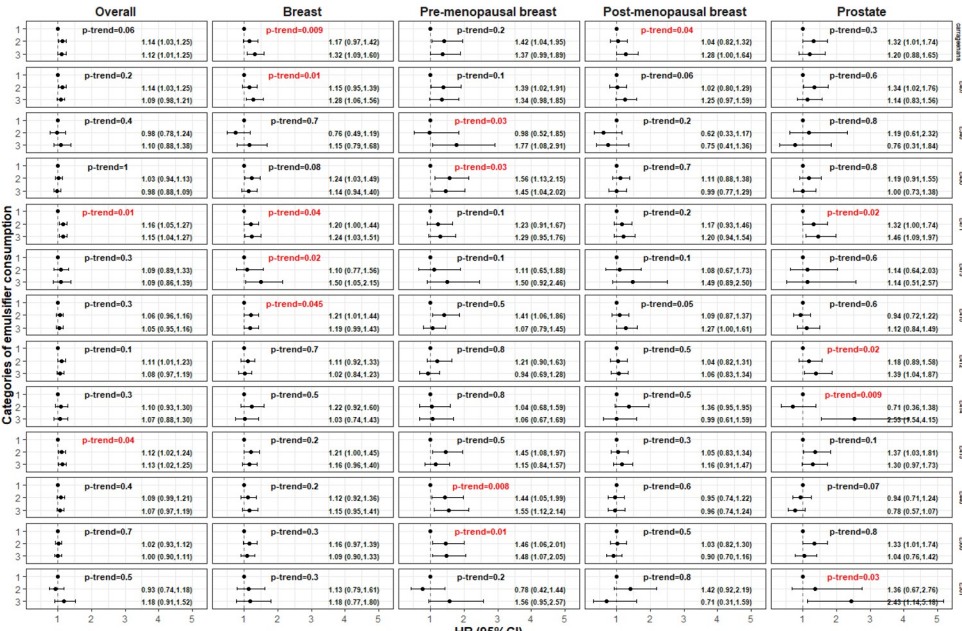

**Fig 4. Associations between selected emulsifier intakes and cancer risks among study participants from the NutriNet-Santé cohort, 2009–2021 ($N$ = 92,000).[a,b]** DAG, diglyceride of fatty acid; FAs, fatty acids; HR, hazard ratio; MAG, monoglyceride of fatty acid. [a]Emulsifiers with at least 1 statistically significant association with cancer risk are represented here. The detail of all investigated associations between emulsifier intakes and cancer risk with corresponding HRs and 95% CIs is provided in eTable D, as well as cut-offs for categories of emulsifier intakes, and number of cancer cases per category of emulsifier intakes. Mean values for emulsifier intake within each category is presented in eTable B. Groups of emulsifiers were defined as follows (European codes): total carrageenans (E407, E407a). The following emulsifiers were coded as sex-specific tertiles: total carrageenans, E407, E412, E415, E440, E450, E471, and E500. Due to a higher proportion of non-consumers among the included participants, the following emulsifiers were coded as non-consumers (first category), low consumers (second category), and high consumers (third category), with low- and high-consumptions defined according to sex-specific median intakes among consumers: E340, E410, E414, E475, and E901. [b]Multivariable Cox proportional hazard models were adjusted for age (time-scale), sex, BMI (continuous, kg/m$^2$), height (continuous, cm), physical activity (categorical IPAQ variable: high, moderate, low), smoking status (never smoked, former smoker, current smokers), number of smoked cigarettes in pack-years (continuous), educational level (less than high school degree, <2 y after high school degree, ≥2 y after high school degree), number of dietary records (continuous), family history of cancer (yes/no), energy intake without alcohol (continuous, kcal/d), daily intakes of alcohol (continuous, g/d), lipids (continuous, g/d), sugars (continuous, g/d), sodium (continuous, g/d), fibre (continuous, g/d), consumption levels of fruits and vegetables (continuous, g/d), red and processed meats (continuous, g/d), and dairy products (continuous, g/d). Breast cancer models were additionally adjusted for oral contraception (yes/no, in total and premenopausal models only), age at menarche (never, <12 y, ≥12 y), number of biological children (continuous), age at first biological child (no child, <30 y, ≥30 y), menopausal status at baseline (premenopausal, postmenopausal, in total models only), hormonal treatment for menopause (yes/no, in total and postmenopausal models only).

were not attained by any of participants, while ADIs for triphosphates (E451, set at 40 mg/kg of body weight/d) [57] were exceeded by <0.1% of participants [17]. These ADIs are theoretically intended to protect consumers against the potential adverse effects of each additive in a given food product. Yet, despite the substantial amount of work on the literature review and the collective expertise performed, evaluations at a given time (and subsequent reference values and regulations) can only be based on the scientific evidence available at that time. Experimental work over the past few years, as well as the present epidemiological study, re-raised the question of the safety of chronic exposure to emulsifier-type additives and the need for re-evaluated ADIs.

To our knowledge, no prior study has investigated the associations between exposures to a wide range of food additive emulsifiers and cancer risk in a large prospective cohort. Despite the lack of epidemiological data on food additive emulsifiers and disease endpoints, a growing body of recent evidence from experimental research has suggested their detrimental impact on health, with a particular interest in polyoxyethylene sorbitan monooleate (E433, also called polysorbate 80) and carboxymethylcellulose (E466), which are non-natural, synthetic compounds derived from oleic acid and cellulose, respectively. These 2 emulsifiers have been suggested to weaken the intestinal barrier and favour microbiota encroachment, in a way that promote chronic intestinal inflammation in animal models [58,59]. Other detrimental impacts of these emulsifiers on inflammation along with alterations to intestinal microbiota diversity and promotion of carcinogenesis have been demonstrated in animal studies [20,26,60,61]. In humans, a first recent randomised controlled trial on carboxymethylcellulose (E466) suggested high supraphysiological intakes (15g/d) over 11 days may promote postprandial abdominal discomfort and markers of gut inflammation, reduce gut microbiota diversity, as well as alter microbiota composition and faecal metabolome. In addition, findings from this randomised controlled trial suggested inter-individual variability of sensitivity to this emulsifier among individuals [25]. Intakes of polyoxyethylene sorbitan monooleate (E433) and carboxymethylcellulose (E466) were much lower in the NutriNet-Santé cohort, which might explain the null association with cancer risk. However, the intestinal pro-inflammatory properties of food additive emulsifiers have been suggested beyond polyoxyethylene sorbitan monooleate (E433) and carboxymethylcellulose (E466), and may extend to carob bean gum (E410) and xanthan gum (E415) [19], along with mono- and diglycerides of FAs (E471), acetic acid esters of mono- and diglycerides of FAs (E472a), and sodium stearoyl-2-lactylate (E481) [18], which might mechanistically explain the observed associations in our study. While most of the available evidence on food additive emulsifiers focuses on gut health, it can be hypothesised that disruptions in the gut microbiota and increased gut inflammation may contribute to a more systemic, low-grade inflammation which may impact other organs [62]. However, further human intervention trials, epidemiological, and preclinical studies, investigating a wide range of largely consumed emulsifiers, are required to elucidate the potential underlying mechanisms by which food additive emulsifiers may promote systemic inflammation and carcinogenesis.

In the case of some emulsifiers, nonlinear relationships were suggested by our models. This might be linked, for example, to saturation of receptors, such as the Toll-like receptor 4, mediator of intestinal inflammation that could be induced by carrageenan exposure [63]. As the exposure to emulsifiers increases, the receptors may become saturated, limiting the emulsifier's ability to exert further effects, leading to a plateau in the response.

Some additive emulsifier substances can also be naturally occurring, such as pectins and celluloses, which are also fibre. Although this might seem counterintuitive given the protective role of fibre on cancer [64], the additive form might exert deleterious health effects, due to the disruption of food matrix in industrial products containing added emulsifiers compared to plants and fruit, which might lead to different effects on human health.

The clinical implications of this study would be that participants having a higher exposure level to several types/classes of emulsifier additives may have a higher absolute risk of developing cancer, compared to those having lower exposure levels. For example, in this study, the absolute risk of developing breast cancer (as a first incident cancer) for a woman aged 60 was 4.1% in the none-to-low exposure to carrageenan emulsifiers category, 4.6% in the medium exposure category, and 5.2% in the third (highest) exposure category. Cancer is a multifactorial pathology, thus as expected, one specific nutritional factor (here, exposure to an emulsifier) does not drastically increase absolute risks per se. However, these results are of high relevance

for citizens, health professionals, and public health stakeholders since these additives, despite being non-essential for human health, are widespread on the global market. Thus, if causality is established, these increased risks may represent substantial numbers of avoidable cases at the population level.

Strengths of this study pertained to its large sample size, prospective design, along with the detailed information on exposures to food additive emulsifiers. Indeed, the NutriNet-Santé reached unique accuracy in the assessment of qualitative and quantitative food additive intakes thanks to detailed and repeated 24-h dietary records, links to multiple food composition databases (OQALI [40], Open Food Facts [16], GNPD [41], EFSA, and GSFA [42]), ad hoc laboratory assays, and dynamic matching to account for reformulations of industrial food items over time [17]. In addition, multiple sensitivity analyses provided similar results, adding to the consistency and robustness of the observed associations.

However, some limitations must be acknowledged. First, even though dietary records were validated against blood and urinary biomarkers for energy and key nutrients, exposure to emulsifiers has not been validated against blood or urine assays due to lack of specific biomarkers. Second, the higher educational background, imbalance towards women (78.8%), and overall more health conscious behaviours in participants from the NutriNet-Santé cohort compared to the general French population warrant caution in the generalisation of our results. Second, due to the observational design of this study potential residual confounding in the observed associations cannot be entirely ruled out, although this concern has been mostly addressed by the use of multivariable Cox models accounting for a wide range of potential confounders. In addition, some exposures to individual emulsifiers were too low among the included participants, which prevented investigations on their associations with cancer risk. However, all available amounts of emulsifiers consumed were included in the calculation of total and groups of emulsifier intakes. Moreover, measurement errors in food additive emulsifier intakes cannot be entirely ruled out, despite the multisource strategy used for retrieving qualitative and quantitative data on food additive intakes. In particular, intakes might have been underestimated in food items exempt from food labelling (i.e., meat from deli counters, non-homemade retail pastries). Besides, to our knowledge, there is no available food composition database to estimate the food content in naturally occurring emulsifiers, such as lecithins or pectins. Therefore, this study focused on food additive emulsifiers only. Furthermore, the number of cancer cases was limited for some cancer locations, particularly for colorectal cancer, which may have prevented the detection of potential associations. Lastly, as classically observed in nutritional epidemiology studies, a significant subsample of the cohort (17%) was flagged as energy underreporters and were excluded from the final sample. Different reasons might explain this phenomenon, mainly social desirability bias, but also entry errors while declaring their dietary intakes. These participants declare abnormally low and implausible energy intakes in absence of any specific restrictive diets. Indeed, participants were asked whether they were following any caloric restriction; in that case, they were included back in the analyses. This ensured that flagged underreporters have true incoherent reporting, and must be excluded. The prevalence of energy underreports in our study (17%) is in the range of those reported in other similar studies around the world: for example, the prevalence of underreporters ranged between 3% and 20% in the multicentric European EPIC cohort [65], 25.1% in the American NHANES study [66], and 18% in the Norwegian Breast Cancer Screening Program [67]. In the nationally representative INCA 3 study conducted in 2016 by the French Food Safety Agency [68], 18% of adult participants were identified as underreporters using the Black method. Underreporters in our cohort were older and were more inclined to be male and current smokers, to have a higher BMI and alcohol intake, and a lower educational level and monthly income [69]. Although their exclusion may limit the generalizability of the

findings, it was necessary, in order to avoid important exposure classification bias. Lastly, depending on the regularity of consumption of specific emulsifiers and variation across time, the number of dietary records completed by the participants may have an impact on the less robust associations, as suggested by our sensitivity analyses. Consequently, caution is needed in interpreting associations that are not consistent across all sensitivity analyses.

To conclude, this study suggests direct associations between exposures to 7 individual and 3 groups of commonly used food additive emulsifiers and cancer risk in a large prospective cohort of French adults. These results provide novel epidemiological insights into the role of food additive emulsifiers on cancer risk. If confirmed by further epidemiological and experimental research, they could lead to a modification in the regulation of emulsifier use by the food industry, through food policies requiring a modification of the ADI of some emulsifiers, or even restricting the use of others, for better citizen protection. Given the recently established links between ultra-processed food, main dietary source of emulsifiers, and human health, the role of emulsifiers in the development of other long-term noncommunicable diseases should also be explored, through epidemiological research, as well as experimental approaches on humans and animal models whenever feasible. In the meantime, several public health authorities recommend limiting the consumption of foods containing "cosmetic" additives, i.e., not essential for consumer safety [70,71].

## Supporting information

**S1 STROBE Checklist. STROBE-nut: An extension of the STROBE statement for nutritional epidemiology.**
(DOCX)

**S1 Appendix. eFigure A:** Correlations between intakes of food additive emulsifiers among participants from the NutriNet-Santé cohort, 2009–2021 ($n$ = 92,000); **eMethod A:** Method for the identification of underreporters of energy intake; **eMethod B:** Detailed quantitative assessment of emulsifiers; **eMethod C:** Method for multiple imputation of missing values; **eMethod D:** Method for deriving emulsifier patterns by principal component analysis and corresponding factor loadings; **eMethod E:** Sensitivity analyses for the associations between food additive emulsifier intakes and cancer risks; **eMethod F:** Assessment of the proportional hazard assumption in multivariable Cox models using the Schoenfeld residual method; **eFigure B:** Correlations between Schoenfeld residuals and timescale (age, y) from multivariable Cox models between emulsifier intakes and overall, overall breast, premenopausal breast, postmenopausal breast, and prostate cancer risks in participants from the NutriNet-Santé cohort, 2009–2021 ($n$ = 92,000); **eMethod G:** Dose-response analyses using restricted cubic splines; **eFigure C:** Restricted cubic spline plot for the linearity assumption of the association between emulsifier intakes and risks of overall, overall breast, premenopausal breast, postmenopausal breast, and prostate cancers in participants from the NutriNet-Santé cohort, 2009–2021 ($n$ = 92,000); **eFigure D:** Cumulative incidence functions of the association between emulsifier intakes and risks of overall, breast, and prostate cancers, respectively, in the NutriNet-Santé cohort using Fine–Gray models, 2009–2021 ($n$ = 92,000); **eTable A:** Detailed contribution of 24 food groups to emulsifier intakes among participants from the NutriNet-Santé cohort, 2009–2021 ($n$ = 92,000); **eTable B:** Mean daily emulsifier intakes in mg/d (SD) among study participants from the NutriNet-Santé cohort, 2009–2021 ($N$ = 92,000); **eTable C:** Absolute risks of cancer at 60 years old according to categories of emulsifier intakes at the same age, NutriNet-Santé cohort, 2009–2021 ($n$ = 92,000); **eTable D:** Associations between emulsifier intakes and cancer risks among study participants from the NutriNet-Santé cohort, 2009–2021 ($n$ = 92,000); **eTable E:** Sensitivity analyses for the associations between emulsifier intakes and

cancer risks among study participants from the NutriNet-Santé cohort, 2009–2021
($n$ = 92,000); **eTable F:** "Any" versus "none" models for the associations between emulsifier
intakes and cancer risks among study participants from the NutriNet-Santé cohort, 2009–2021
($n$ = 92,000); **eTable G:** Associations between patterns of emulsifier intakes (create with princi-
pal component analysis) and cancer risks among study participants from the NutriNet-Santé
cohort, 2009–2021 ($n$ = 92,000).
(DOCX)

## Acknowledgments

We thank Thi Hong Van Duong, Régis Gatibelza, Jagatjit Mohinder, and Aladi Timera (com-
puter scientists); Julien Allegre, Nathalie Arnault, Laurent Bourhis, and Nicolas Dechamp
(data-manager/statisticians); Merveille Kouam and Paola Yvroud (health event validators);
Maria Gomes (participant support) for their technical contribution to the NutriNet-Santé
study. We thank Prof. Raphaël Porcher (Université Paris Cité) for his help in computing abso-
lute risks. We also warmly thank all the volunteers of the NutriNet-Santé cohort.

This work only reflects the authors' view, and the funders are not responsible for any use
that may be made of the information it contains.

Where authors are identified as personnel of the International Agency for Research on Can-
cer/World Health Organization, the authors alone are responsible for the views expressed in
this article and they do not necessarily represent the decisions, policy, or views of the Interna-
tional Agency for Research on Cancer/World Health Organization.

## Author Contributions

**Conceptualization:** Laury Sellem, Bernard Srour, Mélanie Deschasaux-Tanguy, Mathilde
Touvier.

**Data curation:** Laury Sellem, Bernard Srour, Guillaume Javaux, Fabien Szabo de Edelenyi,
Nathalie Arnault, Cédric Agaësse, Alexandre De Sa, Rebecca Lutchia.

**Formal analysis:** Laury Sellem, Bernard Srour, Guillaume Javaux, Nathalie Arnault.

**Funding acquisition:** Mathilde Touvier.

**Investigation:** Laury Sellem, Bernard Srour, Guillaume Javaux, Nathalie Arnault, Mathilde
Touvier.

**Methodology:** Laury Sellem, Bernard Srour, Guillaume Javaux, Nathalie Druesne-Pecollo,
Younes Esseddik, Mélanie Deschasaux-Tanguy, Mathilde Touvier.

**Project administration:** Nathalie Druesne-Pecollo, Younes Esseddik, Fabien Szabo de Edele-
nyi, Pilar Galan, Serge Hercberg, Mathilde Touvier.

**Software:** Younes Esseddik.

**Supervision:** Mathilde Touvier.

**Validation:** Bernard Srour, Mathilde Touvier.

**Writing – original draft:** Laury Sellem, Bernard Srour.

**Writing – review & editing:** Bernard Srour, Guillaume Javaux, Eloi Chazelas, Benoit Chassa-
ing, Emilie Viennois, Charlotte Debras, Nathalie Druesne-Pecollo, Younes Esseddik, Fabien
Szabo de Edelenyi, Nathalie Arnault, Cédric Agaësse, Alexandre De Sa, Rebecca Lutchia,
Inge Huybrechts, Augustin Scalbert, Fabrice Pierre, Xavier Coumoul, Chantal Julia,

Emmanuelle Kesse-Guyot, Benjamin Allès, Pilar Galan, Serge Hercberg, Mélanie Deschasaux-Tanguy, Mathilde Touvier.

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
