## [Editor Report · Decision Letter 0]

23 Feb 2023

Dear Dr Srour, 

Thank you for submitting your manuscript entitled "Food additive emulsifiers and cancer risk: results from the French prospective NutriNet-Santé cohort" for consideration by PLOS Medicine. I am writing to let you know that we would like to send your submission out for external peer review.

Please re-submit your manuscript within two working days, i.e. by Feb 27 2023 11:59PM.

Kind regards,

Callam Davidson

Associate Editor

PLOS Medicine

---

## [Decision Letter · Decision Letter 1]

3 May 2023

Dear Dr. Srour,

Thank you very much for submitting your manuscript "Food additive emulsifiers and cancer risk: results from the French prospective NutriNet-Santé cohort" (PMEDICINE-D-23-00431R1) for consideration at PLOS Medicine. 

[LINK]

In light of these reviews, we will not be able to accept the manuscript for publication in the journal in its current form, but we would like to invite you to submit a revised version that addresses the reviewers' and editors' comments fully. You will appreciate that we cannot make a decision about publication until we have seen the revised manuscript and your response, and we expect to seek re-review by one or more of the reviewers. 

We expect to receive your revised manuscript by May 24 2023 11:59PM. Please email us (plosmedicine@plos.org) if you have any questions or concerns.

We look forward to receiving your revised manuscript. 

Sincerely,

Katrien Janin, PhD

PLOS Medicine

plosmedicine.org

Your manuscript has been assessed by three reviewers whose reports can be found below. As you will see from the comments, the reviewers have raised a number of concerns that need addressing. The academic editor also left comments. Please carefully revise the manuscript to address all comments raised.

We like to add for your consideration to examine ‘any’ vs ‘none’ comparisons, and provide cumulative counts and principal component analysis (dimension reduction).

COHORT STUDIES:

Please ensure that the study is reported according to the STROBE guideline, and include the completed STROBE checklist as Supporting Information. Please add the following statement, or similar, to the Methods: ""This study is reported as per the Strengthening the Reporting of Observational Studies in Epidemiology (STROBE) guideline (S1 Checklist).

Did your study have a prospective protocol or analysis plan? Please state this (either way) early in the Methods section.

a) If an analysis plan (from your funding proposal, IRB or other ethics committee submission, study protocol, or other planning document written before analyzing the data) was used in designing the study, please include the relevant document with your revised manuscript as a Supporting Information file to be published alongside your study, and cite it in the Methods section. A legend for this file should be included at the end of your manuscript. 

For all observational studies, in the manuscript text, please indicate: (1) the specific hypotheses you intended to test, (2) the analytical methods by which you planned to test them, (3) the analyses you actually performed, and (4) when reported analyses differ from those that were planned, transparent explanations for differences that affect the reliability of the study's results. If a reported analysis was performed based on an interesting but unanticipated pattern in the data, please be clear that the analysis was data-driven.

DATA AVAILABILITY STATEMENT:

PLOS Medicine requires that the de-identified data underlying the specific results in a published article be made available, without restrictions on access, in a public repository or as Supporting Information at the time of article publication, provided it is legal and ethical to do so. Please see the policy at http://journals.plos.org/plosmedicine/s/data-availability and FAQs at http://journals.plos.org/plosmedicine/s/data-availability#loc-faqs-for-data-policy

PLOS defines the “minimal data set” to consist of the data set used to reach the conclusions drawn in the manuscript with related metadata and methods, and any additional data required to replicate the reported study findings in their entirety.

ABSTRACT:

Please structure your abstract using the PLOS Medicine headings: Background, Methods and Findings, Conclusions. 

Please remove all other subheaders.

Abstract Background: 

Provide the context of why the study is important. The final sentence should clearly state the study question.

Abstract Methods and Findings:

Please give further details of the studied cohort.

Please define all abbreviations for the reader at first use including those used for statistical reporting , HR, 95% CI, for example (lines 49-50).

Line 45 - “…cohort (42.1y [14.5], 78.8% female, 2009-2021)…” Please clearly define what the numerical values contained within square parentheses depict. 

Suggest use of commas as opposed to hyphens (as these can be confused with negative values) to separate upper and lower bounds. 

In the last sentence of the Abstract Methods and Findings section, please describe the main limitation(s) of the study's methodology.

Abstract Conclusions:

Please address the study implications without overreaching what can be concluded from the data. 

Please interpret the study based on the results presented in the abstract, emphasizing what is new without overstating your conclusions.

Please avoid assertions of primacy ("These results provide the first epidemiological insights....")

AUTHOR SUMMARY:

Line 72 starting “Experimental…” suggest separate bullet point.

Line 86 - Assertions of primacy.

The E numbers are provided with a space between E and the number. The spaces should be removed throughout (e.g. E331 and not E 331).

INTRODUCTION:

Please address past research and explain the need for and potential importance of your study.

Please indicate whether your study is novel and how you determined that.

Please conclude the Introduction with a clear description of the study question or hypothesis.

For in-text reference callouts, citations placed within square parentheses should precede punctuation as follows, line 90 “…60% in the US and the UK [1], and about 30% in France [2] and throughout Europe [3].” Please check and amend throughout. 

Line 112: space missing after microbiota

METHODS and RESULTS:

Please add the following statement, or similar, to the Methods: "This study is reported as per the Strengthening the Reporting of Observational Studies in Epidemiology (STROBE) guideline (S1 Checklist).”

What is the meaning of the grey highlighted text? Please amend/clarify

Line 155-156 “… and urinary and blood markers”. Please provide more detail.

Line 171 – please define “OQALI and INRAE”

Line 173 - you may wish to include “... Agence nationale de sécurité sanitaire de l'alimentation, de l'environnement et du travail - ANSES)...”

Line 208 - “... ANOVA tests …” please define all abbreviations at first use

Line 230 - “… consumed in g/d … ” please define all abbreviations at first use

Please define all abbreviations at first use, check and amend throughout.

Equally, when reporting statistical information please define all abbreviations at first use – SD, HR, CI and so on - for the reader

Suggest reporting statistical information as follows for clarity for the reader “…HR 1.11; 95%CI [1.02,1.21]; p trend=0.02…” (instead of HR=1.11, 95%CI=1.02-1.21, p-trend=0.02)

We note that you have adjusted each additive for the effect of all the others. To help facilitate transparent data reporting, please also include unadjusted analyses for comparison. This also ties into a comment by reviewer 1, who notes that if most people only consume some of the additives, then the evidence relevant to their choice would come from the unadjusted analysis. 

DISCUSSION:

Please present and organize the Discussion as follows: a short, clear summary of the article's findings; what the study adds to existing research and where and why the results may differ from previous research; strengths and limitations of the study; implications and next steps for research, clinical practice, and/or public policy; one-paragraph conclusion.

Your discussion is largely structured in this way, we suggest you elaborate a bit more on possible next steps and/or public policy and remove assertions of primacy (e.g. see line 375).

DISCLAIMER STATEMENTS:

Please remove all statements apart from acknowledgements, author contributions and abbreviations from the end of the main manuscript and include these only in the relevant parts of the manuscript submission form. They will be complied as metadata

TABLES:

We note that the E numbers are provided with a space between E and the number e.g. E 339 instead of E339. The spaces should be removed throughout (applicable to all tables). 

Please ensure that any and all abbreviations detailed in the tables are clearly defined in the caption/legend for the reader (e.g. see SD for table 1 and 2)

Table 1 - Please report p values as p<0.001 and where higher as p=0.002, for example and not as <.001

FIGURES:

Figure titles and captions: We note that the E numbers are provided with a space between E and the number e.g. E 339 instead of E339. The spaces should be removed throughout. 

Figure 1 - Please convert the pie chart in Figure 1 to a table, or another type of graph. Please consider avoiding the use of green and/or red to make your figures more accessible to those with colour blindness. 

Figure 3: The text and numerical values are very small and rather inaccessible to the reader, please revise. Please clearly define the meaning of the dots and lines in the figure caption.

SUPPORTING INFORMATION:

As above, please include the statistical analysis plan/study protocol and STROBE checklist

Line 145 mentions a web based questionnaire. Please provide a copy of the questionnaire as an SI file.

We note that the clinical trial study was registered first on November 8, 2017, updated on July 26, 2021, and has a continuous enrollment. In accordance with ICMJE requirements, PLOS Medicine requires prospective, public registration of a data sharing plan (as part of clinical trials registration) for all trials that begin enrollment on or after January 1, 2019. Do you have a data sharing plan? If so, please add it as supporting information, if not please clearly state the reasons why not.

eFigure 1:Thank you for including the participant flowchart

eFigure 2: E numbers are provided with a space between E and the number. Please amend.

Comments from the AE:

Confounding matters, but the approach is not much justifiable.

1. Looking at the published paper on the associations of artificial sweeteners with cancer incidence in the same cohort at PLOS Medicine: https://journals.plos.org/plosmedicine/article?id=10.1371/journal.pmed.1003950.

This clearly indicates confounding by artificial sweeteners.

2. The authors adjusted for dietary pattern scores, but those are not justifiable. A better approach was taken in the previously published manuscript .

3. The authors aggregated all emulsifiers except the one assessed as the primary exposure, but makes not much sense.

In addition, in the report of artificial sweeteners, the authors failed to find consistent associations in their sensitivity analysis pertaining to dietary assessment methods. However, at this time the authors did not conduct such a careful analysis.

As such, the authors did the analyses parsimoniously, notable because of my familiarity to the previous paper. A small bias may give meaningful impact to the authors' interpretation because the observed associations were rather small.

The presentation of the main findings needs to be improved, not indicating how the associations could be sensitive to different sets of covariates modelled.

Overall, major revisions are required including re-analyses. 

Comments from the reviewers:

Reviewer #1: See attachment

Michael Dewey

Reviewer #2: Peer review of food additive emulsifiers and cancer risk 

The authors have examined the associations of emulsifiers added to food and risk of cancer in a large cohort of French men and women. The authors have linked dietary intake taken from repeat 24 hour recalls to three food composition databases to determine the amount of a number of different food additives and related intakes with risk of cancer after an average follow-up time of 8.5 years. Overall, the study was well conducted, the exposure was measured well and follow-up for cancer outcomes were thorough. I have a few comments that I would like the authors to address: 

1. I am interested in the exposure - food additives and would like to know more. The authors have used a web-based 24 h dietary recall to assess food consumption. They have reported that this method has been validated. Has the relative validity of the intake of emulsifiers added to foods been compared against that from the diet diaries/records? What is the day-to-day variation in the intake of these food additives. How do the intakes obtained from this study compare to that from any other studies?

2. Was there any other further measurement of diet throughout the follow-up period? If not why and what implications might this have for the results of the study?

3. Dietary intake was collected over a period of two years? Did follow-up of the cohort begin before or after this time. It would seem sensible to begin follow-up after the data collection period ended. 

4. The method of ascertaining outcomes in this study appear to be robust and reliable. Did the study authors obtain any further information about the cancers e.g. the stage and grade? 

5. I am wondering how meaningful the outcome of total cancer is in this population when there are a high proportion of breast cancer and then almost half of the cancers are classified as "other". Going back to the introduction and trying to form some sort of hypothesis with risk of cancer, it would seem that those that affect the digestive tract would be most plausibly linked with intake of additives, but only colorectal cancer is mentioned but was not related to intake of additives (as the authors have stated there are small numbers of cases of colorectal cancer. I am just trying to understand how the hypothesis links to the methods and then to the findings.

6. While the authors have shown a p for trend for the tertiles of additives and hazard rate of cancers, most of the associations don't appear follow a dose-response relation with the hazards of cancer very similar between the second and the third tertile. Some comment on this would be helpful. 

Reviewer #3: Thank you for this opportunity to review this important paper from the NutriNet Sante cohort. Many studies have investigated dietary associations between UPF and health/disease, therefore I am delighted to see an analysis that is able to investigate specific food components rather than only UPFs. NutriNet Sante is one of the only cohort studies that collects dietary data at the food item and brand level, and given emulsifier prevalence is subject to wide brand-related variations, this is crucial to understanding emulsifier/health interactions. Therefore, although the study is (yet another) re-analysis of cohort data, it is in fact highly novel (there are almost no cohort analyses of emulsifiers and health) and rigorous (almost no other cohort could analyse such data). 

However, there are some major limitations that must be addressed, both in how data are analysed and how methods and results are reported.

(1) The methods section of this paper is similar to all other NutriNet Sante papers. Which is of course inevitable and to be expected. However, care should be taken to report methods (or the absence of methods) of relevance to the specific research question in the specific paper, especially in relation to dietary assessment methods.

a. For example, Line 154-156 - "The web-based questionnaires used in the study have been tested and validated against both in-person interviews by trained dietitians, and urinary and blood markers.[34,35]" 

My apologies I am not familiar with the two validation papers, so the following comment may be incorrect, however, I would guess that the validation papers relate to validity in measuring energy, macronutrients, and possibly micronutrients? 

But I imagine the 3x 24 h recall's have not been validated to measure emulsifier exposure (ie the exposure of interest in this paper)? Just because a 24 h recall is accurate at measuring intakes of E, prot, fiber etc, does not mean it is accurate at measuring emulsifier exposure. This generic statement should be heavily nuanced for example - have you validated the method for measuring emulsifier intake? (eg against dietitian assessment, 7-day food diary or a stool biomarker e.g. CG or CMC?). Im not expecting this to be done for the purposes of the revision, but it is essential that nutritional epidemiologists are critical of their dietary assessment methods. So if it is not done, please say explicitly that it has not been validated for measuring emulsifier exposure, and please tone down the assertion that your dietary methods are validated, when they possibly may not be for the exposure of interest in this paper.

b. Line 204 - This study included participants from the NutriNet-Santé cohort who completed at least two 24h-dietary records during their first two years of follow-up/…" Please clarify. In the dietary methods you say that every 6 months 3x 24 h recalls were requested. So in line 204 do you mean they were eligible if they:

i. Conducted 3 x 24 recalls in Jan 2018 and then 3 x 24 h recalls in Jan 2019? (ie that the are eligible if they completed TWO PERIODS of 3x 24-h recalls) OR

ii. Conducted 1 x 24 recall in Jan 2018 and then 1 x 24 h recall in Jan 2019? (ie they are eligible if they completed ONLY TWO random 24 h recalls ever).

If it is (i) then great, but please clarify this as it does not read like this. If it is (ii) then this is a very, very weak methodology. The CV of emulsifier intake has, to my knowledge, not been previously measured but I would imagine it was very, very high due to wide variations in food types and emulsifier types in different foods. Thus emulsifier intake, with a high CV, would require many many days of measurement. Certainly 2x 24h recalls would be wholly inadequate. 

You need only to look at Table 2 to see very low intakes of individual emulsifiers (often less than 20 mg/d!!) and yet with very, very high SD. Therefore, how could 2 x 24 h recalls ever sufficiently accurately measure intake.

If it is (ii) then I would strongly request the group consider more stringent criteria for eligibility, 2 x 24 h recalls really will have very little accuracy. And whilst I understand this is an epidemiological study and data were used to compute tertiles of intake, the method to measure these should be optimised. If it is (ii) then please be more stringent with your cut off for the eligible number of 24-h recalls.

c. You report that on average 5>5 24-h recalls were completed. For the reason stated above, the mean is important but does not give the full picture. Therefore please ALSO report the numbers with only two (which you will hopefully now exclude), three, four, five…. And then more than five. This will show the seriousness of the problem of the raw dietary data collection described above.

d. Case ascertainment - "physician expert committee validated every major health event after reviewing the participants' medical records and collecting additional information from the participants' treating physicians or hospitals". Systems may well be different in different countries, but are you saying that for every patient who reported cancer in the questionnaires, that a physician checked the "hospital medical records" for these? this is how this reads. If so then wonderful!, but I just wanted to check, please clarfy as "medical records" cold mean hospital records, or medical questionnaires, and itisn't clear. Im sure you review hospital records which is great, so lets make sure everyone knows how great this approach is.

(2) Adjustments and data presentation - the series of adjustments performed is, again, somewhat generic and not specific to the requirements of this paper (cancer). Examples include:

a. Why were total lipid intake adjusted, for all cancers? What was your scientific and nutritional rationale? Are their known associations of total lipid intake for all cancers, and each of the cancers analysed?

b. For CRC why was total fiber intake not adjusted for? Fiber has a well described association with lower risk of CRC. Intakes are likely to be lower in those with higher emulsifier intake (although I don't know this), although your data in Table 1 suggest higher fibre in lower emulsifier groups). Adjusting for healthy eating pattern is insufficient to adjust for fiber intake. So fiber must be adjusted for in the CRC model.

I can see a sensitivity analysis has been performed, which includes total fibre, but why is the strongest nutritional associated cancer cause not included in the primary model in the primary paper, rather than in a sensitivity analysis?

How different is the data? (sorry I cant see fig 3 on the pdf it is very low quality, and I currently cannot download the actual PNG file. I would prefer explicit summary of the data for model 1 in the MAIN paper for CRC at least, as it adjusts for fibre. It would only be one extra paragraph. 

c. I think the major tertiles displayed (eg in table 1) were based upon total emulsifier intake. However it would appear that all risks were calculated against intakes of the emulsifier groups or individual emulsifiers only - is that right? (that makes sense as individual emulsifiers have different impacts on health). 

But does it therefore mean that the reader cannot see the actual intakes of each emulsifier group or each individual emulsifier that make up Tertile1, T2 and T3? Maybe this is in Supplementary data (my apologies I cannot currently view this). But if it is not then if the reader is to every interpret the risk of each emsulfier group or each emulsifier, then we need to know the intakes in T1, T2, T3 for each emulsifier group and each emulsifier… sometimes you must be making comparisons of tertiles with very very low intakes, which concerns me around accuracy, but also application. We need to see what these values are for each tertial for each emsulfiier group and each emulsifier.

(3) Emulsifier quantification 177-180 - more detail on the emulsifier quantification are required. This is the central strength of this paper, yet is reported very briefly. If there is insufficient text available then it must be summarised in supplementary methods. 

The three approaches used lead to increasing levels of inaccuracy. It is impossible, as described, to estimate the level of inaccuracy, as it is unclear for how many foods and how many emulsifiers was used for each method. Eg were most estimates foods measured using the very accurate method 1? Or were most using the very inaccurate method 3? Examples of required details include, but are not restricted to:

a. "ad-hoc laboratory assays quantifying additives in specific food items (N=2,677 food additive pairs analysed" - please provide detail

i. does this mean the presence of ALL emulsifiers in 2677 foods? 

ii. What are food-additive pairs? This is crucial information. 

iii. For example, assuming conservative estimate of 5000 UPF/emulsifier-containing foods and >60 emulsifiers, that is 300,000 lab analyses if done completely via method 1 (which is therefore clearly not happened)… it is crucial to report how many were laboratory analysed and for what. Therefore For the above please report:

iv. How many food items were analysed 

v. and for their content of how many different emsulifiers?

b. "doses in generic food categories provided by the European Food Safety Authority (EFSA)" - are you saying that EFSA have provided you with the actual quantities of emulsifier in broad food groups? How did you get this information? This is not available for general release. 

Therefore For the above please report:

i. How many food groups were provided 

ii. and for their content of how many different emulsifiers?

c. For the above, where a food item (eg a cheese sauce) does not have any laboratory data, and so you have to use method 2, can you confirm you would only use "food group" assumptions from EFSA (eg for cheese sauces) ONLY if the specific food item you were calculating (ie the specific brand of cheese sauce) did actually contain emulsifiers in the ingredients list. Otherwise blanket assumptions will result in some food items with no emuslfiiers being coded as containing them… and some food items with emulsifiers being coded as not containing them.

d. "generic doses from the Codex General Standard for Food Additives (GSFA).[41]" 

i. Please expand - a general reader will nt understand this.

ii. Again, for the above please report:

iii. How many food items was this used 

iv. and for their content of how many different emulsifiers?

These should be addressed in the rebuttal and briefly in the manuscript or supplementary information.

e. Table 2 is excellent, thank you for detailed intake data.

(4) Line 165 - under and over-reporting:

a. Is the Black method the Golderg method? If so this is the more commonly used term. If not please clarify.

b. Also, the numbers excluded due to under- and over- reporting should be reported in the main text of the paper.

c. I see from supp figures that >21,000 were excluded due to under-reporting. Given this is based upon(if Goldberg, or summarised by Black) then this should be 2 SD beyond the EI(measured)/BMR(est). If normally distributed this would be 2.5% of the population predicted to be under-reporters, but in fact it is 21,423/129,571 - which is >16% of the population under-reporting. 

I realise you cannot do anything about this, but it really is huge and I ask you to consider moving forward, why your dietary assessment tool may be quite so inaccurate. There is no evidence that 16% of the population intentionally under-report, so much of this is likely the inaccuracy of th 24h recalls. 

d. Supp fig 1 implies there were no over-reporter. Which is very odd. Is this actually correct. Again, please consider within your group the accuracy of your data collection tool.

MINOR

(5) The abstract is poorly written - a shame for such a good study. It will be brilliant once ou make improvements, not limited to:

a. The first sentence of the abstract is: "Objective: This study aimed to investigate THESE associations in a large population-based prospective cohort." Given this will be the first line on a paper "these" is meaningless. Please change.

b. "Main outcome measures: Associations with incident cancer risk were assessed using multivariable Cox models" sentences are incomplete - association of cancer with what? How was incidence of cancer measured? (it is not mentioned in the abstract). This is the most detailed every analysis of emsulfiers and disease. Please show that early on.

c. Pleas report average years of follow-up. Crucial in epidemiology but very low in some reports (but not here).

d. Conclusion "In this large prospective cohort, we observed direct associations between cancer risks (overall, breast, and colorectal) and…" no mention of CRC and risk are presented in the abstract.

(6) E number nomenclature. In some places the E numbers are provided with a space between E and the number e.g. E 331 (see author summary - but also check throughout). The spaces should be removed throughout.

(7) Line 100 "…dozens of food additive emulsifiers are found in thousands of daily-used processed food items (e.g. chocolate, pastries, but also ready-to-eat prepared fruits and vegetables). This is confusing to interpret. As written it sounds as though you are saying there are some foods with dozens of emulsifiers in them - which may grab incorrect media headlines. 

Are you saying there are dozens of emulsifiers in the food chain? If so perhaps be more explicit e.g. The numbers of emulsifiers varies in the food chain depending upon national definitions used but can range from XXX in EU to XXX in US". Your own work and that of other national evaluations could be used.

(8) Line 113 "…potentially leading to higher risks of gut diseases such as inflammatory bowel disease[23] and Crohn's disease.[24]" Crohn's is an IBD, therefore please rephrase (e.g. including…)

(9) Human studies:

a. Line 114-116, and Line 327-329. Its probably important to STRONGLY emphasise, in the interests of balance and scientific rigour, that the CMC doses used in this first in human study are super-physiological. For example you show intakes of CMC in your population are 3.9 mg/d, whereas the human study for which you emphasise changes in microbiome and inflammation used 15 g/d!! (almost 4000 times more)

b. Discussion - Line 327 - states that a human study showed high doses of CMC impact microbiome - stating the dose was 15 mg/d. This is 3 log out!

(10) Discussion - the association of celluloses and breast cancer is interesting. Celluloses are also fiber and, although not all are included in the diet from plant foods, cellulose is widely consumed as part of the plant cell wall. Some comment should be added in the discussion about this apparently counter intuitive finding.

[LINK]

---

## [Decision Letter · Decision Letter 2]

9 Oct 2023

Dear Dr. Srour,

Thank you very much for submitting your manuscript " Food additive emulsifiers and cancer risk: results from the French prospective NutriNet-Santé cohort " (PMEDICINE-D-23-00431R2) for consideration at PLOS Medicine. 

[LINK]

In light of these reviews, I am afraid that we will not be able to accept the manuscript for publication in the journal in its current form, but we would like to consider a revised version that addresses the reviewers', editors', and academic editors' comments. 

In revising the manuscript for further consideration, your revisions should address the specific points made.

Please also check the guidelines for revised papers at http://journals.plos.org/plosmedicine/s/revising-your-manuscript for any that apply to your paper. In your rebuttal letter you should indicate your response to the reviewers' and editors' comments, the changes you have made in the manuscript, and include either an excerpt of the revised text or the location (eg: page and line number) where each change can be found. Please submit a clean version of the paper as the main article file; a version with changes marked should be uploaded as a marked up manuscript.

We expect to receive your revised manuscript by Oct 30 2023 11:59PM. Please email us (plosmedicine@plos.org) if you have any questions or concerns.

We look forward to receiving your revised manuscript. 

Sincerely,

Katrien Janin, PhD

PLOS Medicine

plosmedicine.org

Academic Editor Comments: 

The abstract does not reflect the study findings well, and the current highlight seems partly misleading. The authors should revise text on their interpretations.

The authors conducted sensitivity analyses using repeated measures of their dietary data. First of all, the use of repeated measures in a longitudinal study would be not uncommon. Many cohorts with repeated measures accounted for them in their longitudinal analyses, such as Framingham Heart Study, Nurses’ Health Study, and Health Professionals Follow-up Study. Different numbers of repeats and different timing would not be problematic unless meaningful evidence for bias was present.

While detailed modelling approach was unclear and could be improved, the results in “model 4” in eTable 4 are valuable. 

Then, the authors presented some results apparently inconsistent with the results highlighted in the abstract.

For example, the association of E415 with overall cancer showed inconsistent HRs (95% CI): 1.13 (1.02,1.25) in the abstract, and 0.96 (0.86, 1.06) in model 4 of eTable 4; Polyglycerol esters of FAs (E475) and breast cancer did it, too: 1.50 (1.05,2.15) in the abstract, and 1.24 (0.91, 1.71) in model 4 of eTable 4. Guar gum (E412) and prostate cancer did it, too: 1.39 ( (1.04, 1.87) in the abstract, and 0.70 (0.44, 1.12) in model 4 of eTable 4. Gum Arabic E414 and prostate cancer: 2.43 (1.14, 5.18) in the abstract, and 1.26 (0.85, 1.86) in model 4 of eTable 4. (etc.)

Those showed noticeable inconsistencies, and the inconsistency should be recognised by readers clearly. Despite the apparent failure to confirm the robustness of their findings, the authors said, “Overall main results were similar in sensitivity analyses” (Line 351-352). This interpretation is inappropriate or at least not compatible with the results.

The authors should apply the inevitable revisions as follows at least:

1) The authors should document the inconsistency in one of the secondary analyses in the main text (paragraph before Discussion).

2) The authors should discuss the limitation of this study more elaborately by documenting the internal inconsistency according to the number of repeats to analyse.

3) In the abstract, the authors should highlight the results which were consistent in every sensitivity analysis.

4) In the abstract, the authors should note some findings were not robust in sensitivity analyses.

Editorial Comments:

i) In line with the Academic Editor comments, the results for the sensitivity analysis must be accurately reflected in the manuscript (the results and tables can remain in the SI, but needs to be discussed in the abstract, results and discussion)

ii) Include absolute risk (not just focus on relative risk)

Comments from the reviewers:

Reviewer #1: The authors have addressed my points. WHile checking the reision I noticed that in the supplement, specifically eFigure 4, some of the text is in French. I think most people will recognise the French for cumulative incidence functions and probability but "ensembles de covariables" left me scratching my head.

While the authors are editing the file is it possible to include hyperlinks from the table of contents to the main sections? It is quite a chunky document to page through.

Michael Dewey

Reviewer #2: COmments have been dealt with well. One more minor comment.

1. In Table 1, the missing value rows state "N (%)" but only the N is provided. It would be useful to have the % as well for each of the variables. 

Reviewer #3: Thank you for your detailed and thoughtful response to my comments. An already very good manuscript is now much improved.

[LINK]

---

## [Decision Letter · Decision Letter 3]

20 Dec 2023

Dear Dr Srour, 

On behalf of my colleagues and the Academic Editor, I thank you for adding the absolute risks to your manuscript and responding comprehensively to the editorial questions we mailed you separately. 

I am pleased to inform you that we have agreed to publish your manuscript "Food additive emulsifiers and cancer risk: results from the French prospective NutriNet-Santé cohort" (PMEDICINE-D-23-00431R3) in PLOS Medicine.

PRESS

Sincerely, 

Katrien G. Janin, PhD 

Senior Editor 

PLOS Medicine